# Syntheses, Crystal and Electronic Structures of Rhodium and Iridium Pyridine Di-Imine Complexes with O- and S-Donor Ligands: (Hydroxido, Methoxido and Thiolato)

Michel Stephan, Max Völker, Matthias Schreyer and Peter Burger *

Institute of Inorganic and Applied Chemistry, Department of Chemistry, University of Hamburg, Martin-Luther-King-Platz 6, 20146 Hamburg, Germany; michel.stephan@uni-hamburg.de (M.S.); max.johannes.voelker@uni-hamburg.de (M.V.); matthias.schreyer@gmx.de (M.S.)
* Correspondence: burger@chemie.uni-hamburg.de; Tel.: +49-40-42838-3662

**Abstract:** The syntheses of new neutral square-planar pyridine di-imine rhodium and iridium complexes with O- and S-donor (OH, OR, SH, SMe and SPh) ligands along with analogous cationic compounds are reported. Their crystal and electronic structures are investigated in detail with a focus on the non-innocence/innocence of the PDI ligand. The oxidation states of the metal centers were analyzed by a variety of experimental (XPS and XAS) and theoretical (LOBA, EOS and OSLO) methods. The dπ-pπ interaction between the metal centers and the π-donor ligands was investigated by theoretical methods and revealed the partial multiple-bond character of the M-O,S bonds. Experimental support is provided by a sizable barrier for the rotation about the Ir-S bond in the methyl thiolato complex and confirmed by DFT and LNO-CCSD(T) calculations. This was corroborated by the high Ir-O and Ir-S bond dissociation enthalpies calculated at the PNO-CCSD(T) level.

**Keywords:** metal–ligand dπ-pπ-interaction; Ir-O,S bond dissociation enthalpies; DFT; P/LNO-CCSD(T) calculations; oxidation state analysis; metal–ligand charge transfer; non-innocent ligand; pyridine di-imine ligand

## 1. Foreword

Contributing to the special issue commemorating the 150-year legacy of Justus von Liebig is a great honor for us. Liebig's contributions to all fields of chemistry established him as a true general chemist. Although not as widely known, Liebig had a remarkable friendship and collaboration with Friedrich Wöhler, and they shared a common interest in inorganic chemistry [1]. Together with Auguste Laurent and Friedrich Wöhler, Liebig developed the "Radikaltheorie" to explain the structure of organic molecules [2], and his invention of elemental analysis made it possible to determine their constitution [3].

The correspondence author of this publication is an inorganic chemist who obtained his PhD under the guidance of Hans-Herbert Brintzinger at the University of Konstanz. Brintzinger himself was a PhD student in Basel under the mentorship of Hans Erlenmeyer, who was part of the Erlenmeyer line of chemists. Emil Erlenmeyer, in turn, received his PhD from Justus Liebig, which instilled a deep respect for Liebig's contributions to the field of chemistry in all of us [4].

In this publication, we will summarize our research on the synthesis, structure, and reactivity of rhodium and iridium pyridine di-imine (PDI) complexes with alkoxide and thiolate ligands. We combine theory and experiment, as exemplified by Liebig, to address the reactivity of these complexes, revealing new insights into their structure and reactivity. We hope that this publication will contribute to the ongoing legacy of Justus von Liebig and inspire further research in the fields of inorganic and general chemistry.

## 2. Introduction

Low-valent, late-transition-metal complexes with π-donor ligands, e.g., anionic amido, alkoxido, thiolato and fluorido, $-NR_2$, -OR, -SR and -F groups present a special class of compounds. Their study is primarily motivated by their anticipated unique properties and reactivities based on the HSAB principle for these mismatched soft–hard metal–ligand couples. Furthermore, 4-electron-2-orbitaldestabilizing dπ-pπ orbital interactions between occupied d-orbitals on the metal center and ligand-based lone pairs with π-symmetry might lead to a weakening of the bonds and hence higher reactivity. The chemistry of these types of complexes was summarized in several review articles [5–7]; the unique bonding properties and strengths of the M-OR, M-SR and M-F bonds were also addressed by several authors [8–12]. A particular focus was placed on alkoxido ligands with β-hydrogen atoms, e.g., the methoxido group. Initiated by a β-hydrogen elimination step, these complexes were frequently found to be rather unstable [11,13].

Our group investigates square-planar complexes with pyridine di-imine NNN-donors, of which we and others have demonstrated the ability to behave as non-innocent ligands and reasonable π-acceptors [14–20]. Their unique steric and electronic properties enable the stabilization of highly reactive compounds, such as rhodium and iridium methyl and terminal nitrido complexes, capable of intra- and intermolecular C-H, H-H, Si-H and even C-C activation in ferrocene [21–25]. For the synthesis of these methyl and nitrido complexes, using methoxido ligands as starting materials played a crucial role. We noted that the particular thermal stability of the rhodium and iridium OMe unit, which is resistant to β-hydride elimination even at elevated temperatures, was due to push–pull π-interactions between the oxygen π-donor and the PDI π-acceptor [20,26].

In this paper, we report the synthesis, spectroscopic and crystallographic characterization of rhodium and iridium PDI complexes with O- and S-donors. Theoretical methods are employed to provide insight into their electronic structure with a focus on M-O,S dπ-pπ interactions.

## 3. Materials and Methods

### 3.1. Syntheses

3.1.1. Synthesis of Ligand **2**

Tridentate pyridine di-imine NNN-donor ligands are ubiquitous and employed in main group and transition metal complexes, including both, mono- and dinuclear systems [27–31]. They are commonly prepared by the Schiff-base condensation of a 2,6-diketo or dialdehyde pyridine with two equivalents of a desired aniline or amine derivative [27,28]. The most common precursor is 2,6-diacetyl pyridine, which was also mostly employed by us [27,28]. The ketimine methyl groups in these ligands are rather acidic, and the complexes can be deprotonated under basic conditions [32]. We therefore switched to the corresponding phenyl and 4-t-butyl-phenyl imine-substituted alternatives and selected N-aryl groups with 2,6-di-isopropyl substituent groups to increase the solubility of the complexes in non-polar solvents. The synthesis of 2,6-dibenzoyl pyridine starting material and corresponding PDI ligand is reported in the literature [33]. For the preparation of the 4-t-butyl phenyl analogue, the required diketo pyridine derivative **1** was obtained in analytically pure form at 43% yield from the reaction of 2 equiv. 4-tert-butylphenyllithium [34] with N2,N2,N6,N6-tetramethylpyridine-2,6-dicarboxamide [35] (see Supplementary Materials). Condensation with 2,6-di-isopropyl aniline under acid-catalyzed Dean–Stark conditions in toluene then provided the new ligand **2** at 78% yield as yellow crystals (details see Supplementary Materials).

3.1.2. Synthesis of the Rhodium and Iridium Chlorido Complexes

For the syntheses of the rhodium and iridium chlorido complexes, we followed our previously established route [20,22,26] by the reaction of the di-μ-chlorido bridged tetraethylene dimetal precursor [(Rh,Ir(μ-Cl(ethylene)$_2$)$_2$] with $\frac{1}{2}$ equiv. of the PDI ligand

in THF at RT (Scheme 1). The green rhodium and iridium chlorido complexes **4** and **5** were thus obtained in moderate to good yields (70–90%).

**Scheme 1.** Synthesis of the chlorido, hydroxido and methoxido complexes.

The observed sharp signals in the $^1H$ and $^{13}C$ NMR spectra signaled diamagnetic complexes as expected for these $d^8$-configured square-planar systems. The presence of doublets and triplets for the homotopic meta and para protons of the pyridine ring in the $^1H$ NMR spectra were consistent with the (time-averaged) $C_{2v}$-symmetry of the chlorido complexes, which was further corroborated by the expected number of $^{13}C$ NMR resonances (see Supplementary Materials).

### 3.1.3. Synthesis of the Rhodium and Iridium Hydroxido and Methoxido Complexes

The hydroxido (**6**, **7**) and methoxido complexes (**8**, **9**) were prepared following our previously established route by salt metathesis of the corresponding chlorido compounds with excess cesium hydroxide or sodium methoxide in THF (Scheme 1) [20,26]. The green hydroxido and methoxido complexes were readily isolated by extraction with toluene due to their higher solubility. Upon crystallization, the products were available in moderate to excellent yields. The methoxido complexes are highly water-sensitive, which can be employed to obtain the hydroxido compounds by reaction with water in quantitative yield (see Supplementary Materials). The constitution of these complexes was unambiguously established by elemental analysis and/or X-ray crystallography. While the X-ray crystal structures revealed $C_s$-symmetry of the planar $N_3$-M-O-Me,H core units with bent $sp^2$-hybridized oxygen atoms, the NMR spectra clearly evidenced time-averaged $C_{2v}$-symmetry in solution. This is exemplified by the observation of sharp signals for the pyridine para and meta protons, which appeared as a triplet and doublet in a 1:2 integration ratio for complexes **6–9**. Further support was provided by the fully assigned $^{13}C$ NMR spectra of the hydroxido and methoxido compounds, which revealed three resonances for the ortho, meta and para pyridine carbon atoms and one signal for the carbon atom of the ketimine unit. This is consistent with our previous results for the analogous complexes with methyl-substituted ketimines, for which time-averaged $C_{2v}$-symmetry was also revealed through

[1]H and [13]C NMR spectra [20,26]. The compounds are readily soluble in aromatic solvents, pentane and ethers, and readily decompose in dichloromethane to chlorido complexes.

### 3.1.4. Synthesis of Cationic Rhodium and Iridium PDI Complexes

The cationic complexes **10**–**13** were prepared in good to excellent yields by protonation of the hydroxido or methoxido complexes **7**–**9** using acids with non- or weakly coordinating counterions (Scheme 2). Depending on the conditions and starting material, either the cationic aqua (**12**), methanol (**11**) or THF (**10**, **13**) square-planar diamagnetic complexes were obtained. The [1]H NMR coordination chemical shifts of the latter ligands in solution along with sharp resonances and X-ray crystallography in the solid state revealed that the counterions were not coordinated in any case. In dichloromethane, the cationic complexes were almost instantaneously converted to the neutral chlorido compounds **4** or **5**; in fluorobenzene, however, they were stable and dissolved well. In THF solution, the solvent occasionally polymerizes, which can be prevented by the use of the related cyclic ether, 2-methyl tetrahydrofuran.

**7** (M=Ir, R[1]=H, R[2]=H)
**8** (M=Rh, R[1]=tBu, R[2]=Me)
**9** (M=Ir, R[1]=H, R[2]=Me)

**10** 100 % (M=Rh, R₁=tBu, R[3]=THF)
**11** 99 % (M=Ir, R[1]=H, R₃=MeOH)
**12** 93 % (M=Ir, R[1]=H, R[3]=$H_2O$)

**Scheme 2.** Synthesis of the cationic complexes.

### 3.1.5. Synthesis of the Iridium Thiolato Complexes

For the synthesis of the iridium thiolato compounds **14**–**16**, we followed the established routes reported in the literature [36,37]. For the methyl and phenyl sulfido compounds **14** and **15**, salt metathesis with NaSR (R = Me, Ph) in the chlorido complexes was employed (Scheme 3). The hydrosulfido compound **16**, PDI-Ir-SH, was obtained by the reaction of a commercial $H_2S$ solution in THF with the methoxido complex **9**, which was related to the aforementioned reaction of **8** with the weaker acid water (Scheme 2). Most noticeable is the violet color of these compounds, which deviates strongly from the green (green–brown) color observed for all the other square-planar Rh(I) and Ir(I) PDI complexes previously studied by our group. All of the new sulfido complexes were obtained as diamagnetic crystalline solids in moderate to excellent yields. The proposed constitution based on NMR spectroscopy was unambiguously confirmed by X-ray crystallography (vide infra). As will be discussed below, the observed idealized $C_s$-symmetrical structure of the PDI-Ir-SMe core unit in the crystal structure of **14** can also be witnessed in solution by temperature-dependent [1]H NMR spectroscopy (vide infra).

**Scheme 3.** Synthesis of the methyl, phenyl thiolato and sulfido complexes.

### 3.1.6. Synthesis of the Iridium Methyl Complex 18

For comparison, we also synthesized and crystallographically characterized the PDI iridium methyl complex **18**, which carries methyl rather than aryl groups at the ketimine carbon atom. This diamagnetic compound was prepared from the chlorido complex by reaction with dimethyl zinc.

### 3.2. Methods
### 3.2.1. Theoretical Methods
#### DFT Calculations

For the geometry optimizations of the ground and transition states, we employed DFT calculations with the PBE functional [38], including dispersion corrections by Grimme's D3 method [39] with Becke–Johnson damping (D3BJ) [40]. Def2-TZVP basis sets were employed for all atoms [41]. For rhodium and iridium, def2-ECP pseudopotentials were used (Ir: ECP-60-MWB, Rh: ECP-28-MWB) [42]. The RI-DFT method [43] was used with the corresponding RIJ-auxiliary basis [44]. For the PW6B95 hybrid functional [45], semi-numeric exchange ($senex keyword in Turbomole) was employed [46]. Solvation effects were included within the COSMO formalism [47] using a dielectric constant of $\varepsilon = 7.6$ for THF. The geometries were fully optimized without geometry or symmetry constraints; minima were confirmed by the absence of imaginary frequencies in the calculations of the analytic second derivatives; for the transition states, only one imaginary frequency was observed. Transition state optimizations were carried out with Kästners DL-FIND optimizer [48] implemented in TCL-Chemshell 3.7 [49] starting from the transition state geometries obtained from linear transit searches. IRC calculations were carried out to confirm that the transition states connected the starting material and product. The calculations were carried out with version 7.7.1 of the parallelized Turbomole program package [50] on our local 32-core and 96-core machines equipped with 512 GB and 3 TB RAM, respectively, and the two 40-core nodes (1 TB RAM) of the "Hummel" computing cluster of the University of Hamburg computing center (RRZ) employing fast SSD/NVME scratch disk space.

#### Local Coupled Cluster Calculations

For the closed-shell systems, local natural orbital LNO-CCSD(T) calculations [51] were carried out with the freely available MRCC (2022) program package using the default thresholds (lcorthr = normal) [52]. Geometries optimized at the PBE-D3BJ/def2-TZVP level and def2-TZVPP basis sets in combination with complementary def2-TZVPP/C auxiliary correlation basis [53,54] sets and def2-TZVP pseudopotentials [42] were employed. For reference, density-fitted Hartree–Fock calculations def2-QZVPP/JK auxiliary basis were used [55]. The solvation correction was obtained from the energy differences of two single-point calculations at the PBE-D3BJ(COSMO($\varepsilon = 7.6$)/def2-TZVP) and PBE-D3BJ/def2-TZVP (gas phase) levels. Back-corrections for the LNO-CCSD(T) energies to free enthalpies ($\Delta G_{298}$) were carried out with thermochemical data obtained from the DFT calculations

at the PBE-D3BJ/def2-TZVP level. A value of 1.011 was taken as the scaling factor from Truhlars database (ver. 5.0) [56]. The values for the T1 and D1 diagnostics were typically T1 $\approx$ 0.015 and D1 $\approx$ 0.15, signaling single reference cases.

The evaluation of bond dissociation enthalpies requires local couple cluster calculations of open-shell (S = 1/2) systems. Since this feature is currently not available in MRCC, we switched to PNO-U(R)-CCSD(T1) calculations [57] with Molpro version 2022.3 with the domopt = tight setting [58]. The SO-SCI SCF optimization scheme was employed to converge to the ground state of the Hartree–Fock reference wave function. Enthalpy corrections of the thermochemical data were provided by the "freeh" program of the Turbomole package from the analytical second derivatives obtained at the UPBE-D3BJ/def2-TZVP level. A value of 1.011 was taken as the scaling factor from Truhlars database (ver. 5.0). For single atoms (H, Cl), a value of 5/2 RT was used. The default def2-TZVP pseudopotentials were employed for rhodium (ECP-28MWB) and iridium (ECP-60MWB). For all atoms, the corresponding def2-MP2FIT auxiliary density-fitting basis was used [54]. The values for the T1 and D1 diagnostics were typically T1 $\approx$ 0.015 and D1 $\approx$ 0.15, signaling single reference cases, which was further corroborated by negligible spin contamination (<S**2> $\approx$ 0.75) of the Hartree–Fock reference wave functions of the S = 1/2 radicals.

Charge Transfer and Electron Decomposition Analysis

ALMO-EDA & ALMO-CTA: Energy decomposition analysis (EDA) [59] and charge transfer analysis (CTA) [60] are methods for examining interactions within a chemical system involving fragments [59]. It allows for the decomposition of the overall interaction energy into distinct contributions, e.g., permanent electrostatics, Pauli repulsion, dispersion, and charge transfer. EDA and CTA approaches are applied with great success in theoretical chemical investigations of electronic structures. A recent variant is ALMO-EDA [61] implemented in the Qchem program package [62], which builds on absolutely localized molecular orbitals and enables the analysis of covalently bonded fragments. For our study, we were particularly interested in the charge transfer to and from the PDI ligand fragment, which was analyzed with regard to the energetics (in kJ/mol) and transferred electrons (in me$^-$). The most important contributions were identified by the corresponding localized orbitals called complementary occupied virtual pairs (COVPs). ALMO-EDA/CTA calculations were performed with Qchem 6.0 for converged WB97X-D [63] Kohn–Sham wave functions (def2-TZVP basis and def2-ECP pseudopotentials for rhodium and iridium).

QTAIM analysis: The topology analysis method, a component of quantum theory of atoms in molecules (QTAIM), studies electron density gradients ($\rho$) with a focus on critical points (CPs), where these gradients are zero ($\nabla\rho(CP) = 0$) [64]. CPs are further characterized by negative eigenvalues ($\lambda_1$, $\lambda_2$, $\lambda_3$) of the Hessian matrix. We explored bond critical points (termed (3, −1)) with two negative eigenvalues, located on bond paths connecting local maxima in all directions ((3, −3) critical points). The latter aligned closely with nuclear positions. The size of $\nabla\rho(BCP)$ and the sign of $\nabla^2\rho(BCP)$ distinguish interaction types (covalent, ionic, and van der Waals). Bond ellipticity ($\varepsilon$), defined as $\varepsilon = \lambda_1/\lambda_2 - 1$ ($|\lambda_1| \geq |\lambda_2|$), reveals bonding types. For $\varepsilon = 0$, like cylindrically symmetrical distributions, examples include the single bond in ethane and the triple bond in acetylene. $\varepsilon$ serves as a measure for $\pi$ double bonding, reaching 0.45 for ethylene. QTAIM defines atom basins, within which electronic density integration provides the atomic charge. For the charge transfer analysis, the aggregated charges of the atoms of the PDI ligand fragment were used. The QTAIM calculations were performed with the MultiWfn (ver. 3.8) program package [65]. The required wfx file was obtained from a Turbomole-generated Molden file and subsequent conversion by Molden2aim [66]. Both basis sets with ECPs (def2-TZVP) and the x2c-TZVPall all-electron basis sets in combination with the X2C relativistic all-electron approach were tested and gave essentially identical results.

NBO analysis: Natural bond analysis (NBO) is a computational method to analyze the electronic structure and chemical bonding within molecules [67]. It focuses on identifying natural orbitals and Lewis-like orbitals by computing bonding orbitals characterized by

the highest electron density. NBO helps reveal the nature of chemical bonds, lone pairs, and charge transfer interactions, providing valuable insights into a molecule's electronic characteristics and bonding patterns. The NBO 7.0 calculations [68] were performed with a FILE47 input file generated by an Orca 5.04 [69] DFT PBE-D3BJ/def2-TZVP single-point calculation using geometries optimized by Turbomole. Molecular orbital plots (NBO and ALMO-EDA(COVP)) were generated from cube files with Chemcraft ver. 1.80 [70].

Oxidation State Analysis

For the determination of the (non-physical observable) oxidation states of the metal centers, three different theoretical methods—(i) the localized orbital bond analysis (LOBA) [71], (ii) the effective oxidation states (EOS) [72,73] and (iii) the oxidation state localized orbitals (OSLO) analyses—were employed [74]. The LOBA method was previously promoted by Head-Gordon et al., but displays deficiencies for systems with strong electron delocalization [71]. It was superseded by the OSLO methodology by the same group, which helped to solve difficult cases [74]. The OSLO method was recently benchmarked with the effective oxidation state (EOS) analysis [73] developed by Salvador et al., which indicated that both are robust methods to assign oxidation states based on the defined fragments [74]. For the oxidation state analyses, Head-Gordon's WB97X-D functional was employed using either Qchem 6.018 [62] or Gaussian 16 rev. C01 19 [75] with tight DFT grids (def2-TZVP basis and def2-ECP pseudopotentials for rhodium and iridium). Oxidation states based on the LOBA and OSLO method were calculated with a prelease version of Q-Chem 6.1 with either Loewdin or Mulliken charges. The APOST-3D program of Pedro Salvador Sedano [76] was used to perform the calculations of the oxidation states by the EOS method using a Gaussian 16 formatted checkpoint file. QTAIM and TFVC charges were employed.

Bond Order Analysis

The AOMIX program package ver. 6.92 was utilized to decompose the Wiberg/Mayer bond orders into their symmetry components [77–79]. The required Molden input file was prepared by Turbomole with the tm2aomix program for a PBE-D3BJ/def2-TZVP(def2-ECP) calculation of an optimized geometry at that level.

Wieghardt's (Geometric) Analysis of Bonding Metrics

Based on the analysis of a large number of X-ray crystal structures of main group and transition metal PDI complexes, Wieghardt et al. devised a diagnostic tool to assign the reduction state of the PDI ligand, ranging from neutral PDI to $PDI^{-4}$, based on its bonding metrics [16]. For this method, the C-N and C-C distances in the pyridine and ketimine units ($r(C_{py}-N_{py})$, $r(C_{im}-N_{im})$) and the exocyclic C-C bond ($rC_{py}-C_{im}$) between the pyridine and ketimine carbon atoms are condensed into a single parameter, $\Delta_{geo} = r_{avg}(C_{py}-C_{im}) - (r_{avg}(C_{py}-N_{py}) + r_{avg}(C_{im}-N_{im}))/2$ (avg: averaged). Combining the analyzed crystallographic data with other spectroscopic or theoretical findings, Wieghardt et al. defined ranges for $\Delta_{geo}$ corresponding to the reduction state of the PDI ligand. This method relies on the fact that the occupation of $\pi^*$ orbitals in the reduced PDI ligand leads to shortening of the exocyclic C-C bonds and lengthening of the C-N bonds in the pyridine and ketimine groups. It should be noted that Wieghardt primarily utilized 3d metal systems to calibrate the $\Delta_{geo}$ parameter. In comparison to their 4d and 5d congeners, 3d transition metal complexes exhibit a significantly larger variation in the bonding parameter for reduced PDI ligands.

Local Vibrational Mode Analysis

Cremer and Kraka et al. developed the local mode vibrational theory, which allows for the derivation of local vibrational force constants based on the Hessian matrix of a QM calculation [80–82]. The calculations were performed with the LMODEA(F90) program kindly provided by Elfi Kraka [83]. The required Hessians (and dipole gradients) were obtained from Turbomole (aoforce) calculations of the analytical second derivatives.

## 4. Results and Discussion

### 4.1. X-ray Crystal Structures

We were able to determine X-ray crystal structures for most of the compounds reported herein. The most important structural parameters are compiled in Table 1, which includes reference data for related square-planar complexes for comparison. The molecular structures are presented in Figure 1.

**Table 1.** Selected averaged bond distances and angles determined by X-ray crystallography with DFT-optimized values (PBE-D3BJ/def2-TZVP, Rh, Ir, def2-ECP) in parentheses and Wieghardt's $\Delta_{geo}$ parameter.

| Cpd./ Parameter | M: Σ of Angles [°] | $N_{imine}$-M-$N_{imine}$ [°] | M-X-R [°] | $r_{avg}$((PDI)Ir-X) [Å] | $r_{avg}$ (M-$N_{imine}$) [Å] | $r_{avg}$(M-$N_{pyridine}$) [Å] | $r_{avg}$ ($C_{im}$-$N_{im}$) [Å] | $\Delta_{geo}$ [Å] [a] |
|---|---|---|---|---|---|---|---|---|
| IrSPh 15 | 359.89 (360.10) | 157.41 (157.31) | 122.38 (124.03) | 2.251 (2.252) | 2.032 (2.030) | 1.914 (1.929) | 1.332 (1.349) | 0.083 (0.072) |
| IrSH 16 | 359.89 (360.0) | 157.90 (158.53) | (103.54) | 2.269 (2.262) | 2.019 (2.010) | 1.913 (1.922) | 1.338 (1.348) | 0.082 (0.078) |
| IrSMe 14 | 360.04 (359.99) | 157.49 (157.63) | 118.76 (118.82) | 2.241 (2.243) | 2.017 (2.020) | 1.920 (1.931) | 1.340 (1.350) | 0.078 (0.074) |
| IrOMe 9 | 359.95 (360.01) | 158.43 (159.21) | 130.50 (129.96) | 1.968 (1.954) | 2.022 (2.014) | 1.893 (1.896) | 1.337 (1.348) | 0.082 (0.071) |
| [Me]IrOMe 17 | 359.98 | 159.01 | 134.24 | 1.951 | 2.00 | 1.870 | 1.349 | 0.063 |
| RhOMe [d] 8 | 359.96 (360.01) | 158.69 (158.73) | 134.01 (130.39) | 1.942 (1.949) | 2.035 (2.028) | 1.889 (1.897) | 1.327 (1.338) | 0.100 (0.088) |
| IrOH [b] 7 | 359.98 (360.0) | 159.29 (159.95) | (111.16) | 2.027 (1.964) | 2.00 (2.005) | 1.876 (1.892) | 1.335 (1.348) | 0.054 (0.074) |
| RhOH 6 | 359.99 (359.99) | 158.76 (159.64) | (110.47) | 2.012 (1.957) | 2.010 (2.015) | 1.888 (1.892) | 1.325 (1.338) | 0.104 (0.095) |
| IrCl [c] 5 | 359.87 (359.99) | 158.61 (159.88) | | 2.256 (2.309) | 2.015 (2.008) | 1.910 (1.898) | 1.333 (1.343) | 0.088 (0.087) |
| RhCl 4 | 359.95 (359.99) | 157.03 (159.60) | | 2.303 (2.309) | 2.075 (2.017) | 1.883 (1.897) | 1.319 (1.333) | 0.113 (0.102) |
| IrMe 18 | 359.97 (359.85) | 157.54 (157.46) | | 2.115 (2.086) | 2.003 (2.008) | 1.933 (1.949) | 1.333 (1.340) | 0.091 (0.088) |
| Ir(MeOH)+ 11 | 359.96 (359.99) | 159.56 (160.30) | | 2.097 (2.130) | 2.023 (2.012) | 1.891 (1.884) | 1.338 (1.336) | 0.081 (0.113) |
| Ir($H_2O$)+ [d] 12 | 360.00 (360.01) | 160.26 (161.05) | | 2.090 (2.137) | 2.026 (2.002) | 1.873 (1.879) | 1.315 (1.335) | 0.132 (0.106) |
| Ir(THF)+ 13 | 360.18 (360.04) | 158.55 (159.59) | | 2.111 (2.141) | 2.033 (2.026) | 1.883 (1.889) | 1.321 (1.336) | 0.108 (0.099) |
| Rh(THF)+ 10 | 360.14 (360.02) | 158.62 (159.3) | | 2.181 (2.156) | 2.026 (2.036) | 1.901 (1.887) | 1.308 (1.328) | 0.123 (0.111) |
| IrN 19 | 360.1 (360.15) | 150.86 (149.71) | | 1.647 (1.699) | 2.010 (2.031) | 2.003 (2.036) | 1.329 (1.347) | 0.065 (0.054) |
| IrNO 20 | 360.01 (360.01) | 156.55 (155.74) | 176.36 (179.83) | 1.751 (1.783) | 1.983 (2.017) | 1.886 (1.932) | 1.333 (1.354) | 0.037 (0.037) |
| IrNH$_2$ 21 | 360.0 (360.0) | 159.51 (158.57) | | 1.925 (1.938) | 2.002 (2.005) | 1.886 (1.904) | 1.363 (1.346) | 0.036 (0.063) |
| IrCO+ 22 | 360.0 (360.0) | 157.08 (156.82) | 180.0 (180.0) | 1.846 (1.873) | 2.022 (2.028) | 1.980 (1.990) | 1.308 (1.321) | 0.155 (0.135) |

[a] $\Delta_{geo} = r_{avg}(C_{py} - C_{im}) - (r_{avg}(C_{py} - N_{py}) + r_{avg}(C_{im} - N_{im}))/2$; [b] structure of lower quality; [c] disorder not resolved; [d] M and $N_{py,para}$ on special positions ($C_2$-axis).

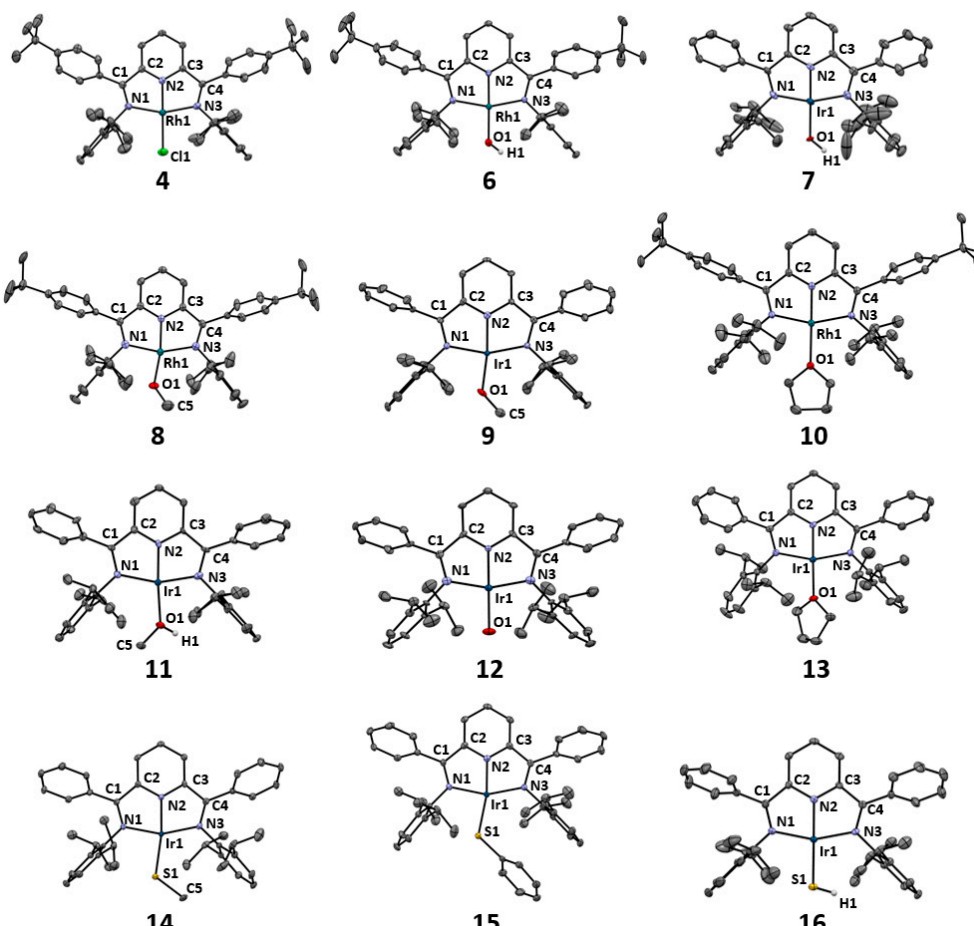

**Figure 1.** Ortep plots of the complexes reported herein with anisotropic displacement parameters shown at the 50% probability level. Hydrogen atoms except for the OH and SH units, counterions and solvent molecules are omitted for clarity.

The sum of the angles around the metal centers was approximately 360° for all compounds, revealing planar coordination geometries. As previously observed for these types of four-coordinate Rh and Ir PDI complexes, the angles between the trans imine nitrogen atoms and the metal centers fell within a narrow range of 157.4–160.3°, which deviated from the ideal 180° in square-planar geometry due to the short imine C-N double bonds [20]. This deviation has significant implications for the interactions of the $d_{xz}$ and $d_{yz}$ orbitals and the $\pi^*$-system of the PDI ligand and is particularly relevant when comparing with PNP–pincer ligands. Notably, the metal–$N_{imine}$ bond distances showed only a negligible variation, with an average of 2.02 ± 0.01 Å. Similarly, the $N_{pyridine}$-metal bond lengths fell within a narrow range of 1.88 to 1.92 Å, despite the variation of the trans influences of the ligands investigated (as compiled in Table 1). The rigidity of these metal PDI geometric parameters was only broken in the carbonyl and nitrido complexes, which exerted the strongest trans influence, leading to Ir-$N_{pyridine}$ distances of 1.980 Å and 2.003 Å. This required a reduction of the $N_{imine}$-Ir-$N_{imine}$ angle to 150.86° in the nitrido complex to maintain a planar structure. In conclusion, all the complexes studied in this work can be assigned to a pseudo-square-planar coordination geometry. For the sake of clarity, further on we will refer to this geometry as square-planar.

The forthcoming analysis of the electronic structures of the complexes described herein will focus on two aspects: (i) the non-innocence of the PDI ligand and the oxidation states of the metal centers, and (ii) the characterization of the Ir-X bond.

### 4.2. Electronic Structure of the Complexes

4.2.1. Non-Innocence of the PDI Ligand

The non-innocence of PDI ligands is well-established and was mostly studied for 3d transition metal systems [14–18]. We have previously analyzed the situation of our square-planar 4d and 5d PDI transition metal complexes for a wide variety of additional fourth ligands ranging from strong $\pi$-donors to $\pi$-acceptors, i.e., nitrido and carbonyl groups [20]. For these investigations, experimental XPS, XAS and $^{13}$C-NMR data were combined with theory (DFT and CASSCF) including a localized orbital analysis (LOBA) to assess the oxidation states of the metal centers [19,20,23]. In contrast to their 3d congeners, in most cases the metal PDI interaction in the Rh and Ir complexes is best described by back-donation to the $\pi$*-acceptor orbital of the PDI ligand. This can be traced to the better metal–ligand $d\pi$–$\pi$* overlap of the larger and more diffuse 4d and 5d orbitals. Nevertheless, it should be noted that the transition between the limiting back-donation and biradical scenarios in non-innocent and innocent PDI ligands is continuous [17]. We identified clear examples with non-innocent, doubly and singly reduced PDI ligands in the nitrido and nitrosyl iridium complexes (($PDI^{2-}$)Ir(III)(N$^-$), ($PDI^{2-}$)Ir(I)(NO$^+$)) and neutral carbonyl and dinitrogen compounds (($PDI^-$)Ir(I)(CO), ($PDI^-$)Rh(I)(N$_2$)). On the other hand, neutral PDI ligands were established for cationic complexes, while the situation for complexes with N and O $\pi$-donors (NRH: R = H, SiR$_3$ and Ph; OR: R = H, Me) was borderline in terms of non-innocent PDI ligands. In the course of the analysis of the hydrogenolysis experiments described herein, we made one more attempt to investigate the electronic situation of the PDI ligands and included new diagnostic tools based on experimental bonding metrics and theory.

To analyze the innocence or non-innocence of the PDI ligand by theoretical methods, we utilized small parent model complexes instead of those with methyl or aryl substituents of the ketimine units. The Absolutely Localized Molecular Orbitals Energy Decomposition Analysis (ALMO-EDA) in Q-Chem [61,84] was employed to assess the charge transfer between the PDI ligand and the M-X units in the (PDI)M-X complexes. Two fragments were defined: the PDI ligand and the M-X units, and their interaction energies and charge transfer were analyzed [61,84]. A further attempt to determine the charge transfer between the PDI ligand and the IrX unit was made by investigating the aggregated QTAIM charges of the PDI ligand atoms (Table 2). Additionally, the oxidation states of the metal centers were assessed by three different theoretical methods: (i) the localized orbital bond analysis (LOBA) [71], (ii) the effective oxidation states (EOS) [72,73] and (iii) the oxidation state localized orbital (OSLO) analyses [74]. Where available, we also included complementary experimental XPS and XAS data. Furthermore, the Wieghardt et al. diagnostic geometrical parameter $\Delta_{geo}$ for the PDI ligands was tabulated for the PDI ligand derived from X-ray crystal structure data along with those from DFT-optimized geometries for the full system in parentheses [16]. To illustrate the variation for the different complexes, we also compiled the experimental (X-ray) and DFT-calculated C-N bond distance between the pyridine and ketimine carbon atoms ($rC_{py}$-$C_{im}$). Finally, the local force constants for this bond in the corresponding model complex obtained by local vibrational mode analysis with the LMODEA program package [80] are compiled in Table 2.

**Table 2.** Analysis of the charge of the PDI ligand, (ALMO-EDA) electron transfer from to the PDI ligand, oxidation states of the metal centers (XPS/XAS, LOBA, EOS, OSLO) and Wieghardt's $\Delta_{geo}$-parameter (theoretical value in parentheses). $\Delta_{CT}$ equals $\Delta_{CT} = CT(PDI \rightarrow MX) - CT(PMX \rightarrow PDI)$ (values in charge units $\Delta Q$ of $1/1000$ of an electron: me$^-$). The non-/innocence of the PDI ligand is indicated by the color-coding (orange: non-innocent; yellow: borderline; green: innocent); for details see text.

| Cpd. M-X/Method | ALMO-EDA | | | QTAIM | Oxidation State Analysis | | | | Wieghardt Analysis | | Local Force |
|---|---|---|---|---|---|---|---|---|---|---|---|
| | $E_{CT}$/CT PDI → MX [kJ/mol]/ [me$^-$] | $E_{CT}$/CT MX → PDI [kJ/mol] /[me$^-$] | $\Delta_{CT}$ [me$^-$] | SPDI Charge [me$^-$] | XPS /XAS | LOBA $^c$ M | EOS M, X, PDI | OSLO M, X, PDI | $\Delta_{geo}$ Parameter [Å] | r(C$_{py}$C$_{im}$) X-ray (calc.) [Å] | ν (C$_{py}$C$_{im}$) [mdyn/Å] |
| **IrNO** | −383/148 | −151/208 | −60 | −627 | +1 | +1 | +1, +1, −2 | +5, −1, −4 * | 0.037 (0.037) | 1.377 (1.403) | 6.103 |
| **IrN** | −348/132 | −128/157 | −25 | −530 | +3 | +3 | +3, −1, −2 $^b$ | +3, −1, −2 | 0.065 (0.054) | 1.424 (1.421) | 5.758 |
| **IrNH$_2$** | −357/126 | −169/180 | −54 | −607 | +1 | +3 | +1, −1, 0 $^a$ | +1, −1, 0 * | 0.036 (0.063) | 1.392 (1.415) | 5.536 |
| **IrOMe** | −368/132 | −153/150 | −18 | −487 | | +3 | +1, −1, 0 | 1, −1, 0 * | 0.082 (0.071) | 1.444 (1.440) | 5.420 |
| **IrMe** | −342/120 | −152/154 | −34 | −469 | | +1 | +1, −1, 0 | 1, 1, −2 * | 0.091 (0.088) | 1.445 (1.445) | 5.170 |
| **IrSMe** | −365/135 | −135/128 | 7 | −464 | | +1 | +1, −1, 0 | 1, −1, 0 * | 0.078 (0.074) | 1.439 (1.441) | 5.426 |
| **RhOMe** | −251/102 | −111/102 | 0 | −376 | | +1 | +1, −1, 0 | 1, −1, 0 * | 0.100 (0.088) | 1.451 (1.447) | 5.300 |
| **IrCl** | −390/146 | −135/117 | 29 | −362 | +1 | +1 | +1, −1, 0 | +1, −1, 0 * | 0.088 (0.087) | 1.438 (1.450) | 5.238 |
| **RhCl** | −273/115 | −96/78 | 37 | −253 | | +1 | +1, −1, 0 | +1, −1, 0 * | 0.113 (0.102) | 1.459 (1.456) | 5.098 |
| **IrTHF$^+$** | −403/156 | −114/91 | 65 | −8 | | +1 | 1, 0, 0 | 1, 0, 0 | 0.108 (0.099) | 1.456 (1.456) | 5.190 |
| **RhTHF$^+$** | −282/125 | −81/60 | 65 | +82 | | +1 | 1, 0, 0 | 1, 0, 0 | 0.123 (0.111) | 1.458 (1.461) | 5.075 |
| **IrCO$^+$** | −406/159 | −67/42 | 117 | +302 | | +1 | +1, 0, 0 | 1, 0, 0 | 0.155 (0.135) | 1.481 (1.472) | 4.590 |

$^a$ +3, −1, −2 (TFVC scheme); $^b$ QTAIM scheme; +1, +1, −2 with Mulliken scheme; +1, +1, −2 (L); $^c$ PBE0 functional; * branching option was used.

### 4.2.2. ALMO-EDA

The ALMO-EDA was facilitated by the inspection of the complementary occupied/virtual pairs of orbitals (COVPs) displaying the most significant contributions between pairs of the interacting fragments. The COVPs for the ligand → metal and metal → ligand interactions are exemplified for the methyl thiolato model complex shown in Figure 2.

Inspection of the COVPs for the PDI → IrSMe charge transfer shown in Figure 2 reveals the expected σ-donation of the ketimine and pyridine N-donors to the metal center. The corresponding π-back-donation to the PDI ligand was reflected in the IrSMe → PDI COVPs. This interaction was weaker and involved the d$_{xz}$ and d$_{yz}$ orbitals with the approximately identical energies of −48.3 and −45.0 kJ/mol, respectively. It deserves special mention that there was also a weaker back-donation from the d$_{xy}$ orbital to a ligand-based orbital. Energetically, the binding was dominated overall by donation, which can be readily seen from the aggregated charge transfer energies of −365 kJ/mol for $E_{CT}$(PDI → IrSMe) and −135 kJ/mol for $E_{CT}$(IrSMe → PDI) (Table 2). This was contrasted by systems with essentially no charge transfer between the fragments, as in the methyl thiolato model complex (PDI → IrSMe: 135 me$^-$ vs. IrS → PDI 128 me$^-$). After this more detailed analysis of the methyl thiolato model complex, we will turn to the data for the other complexes compiled in Table 2.

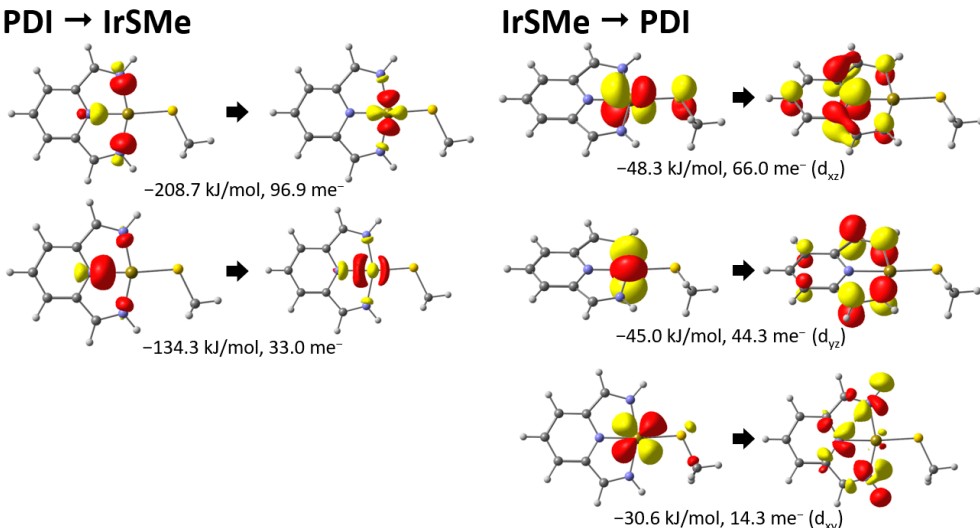

**Figure 2.** Most important COVPs for the methyl thiolato model complex with energy and charge transfer contributions.

Firstly, the ALMO-EDA data clearly established a substantially weaker donation of the PDI ligands in rhodium complexes ($E_{CT}$(PDI → RhX) < 300 kJ/mol) compared to the values for the iridium systems, which ranged from ca. 350 to 400 kJ/mol. This was also observed for the metal to PDI π-back-donation and is readily explained by stronger bonding in the heavier 5d iridium metal centers due to relativistic effects; it is also noted for the M-X bond strengths discussed below (Table 3).

With regard to the charge transfer, the PDI → MX donation showed only a small variation (Ir: 120–159 me$^-$; Rh: 100–125 me$^-$); as expected, smaller values were observed for better donors X and larger ones for the cationic systems. This is in contrast to the charge transfer related to back-donation (MX → PDI), which displayed a strong dependence on the MX unit, ranging from 208 me$^-$ for the nitrosyl ligand to 42 me$^-$ for the IrCO$^+$ fragment in the cationic carbonyl complex.

Overall, the direction of charge transfer was best analyzed by the difference $\Delta_{CT}$ = CT(PDI → MX) − CT(MX → PDI). For the strong donor ligands, X = NO, NH$_2$, N, Me and OMe; negative values for $\Delta_{CT}$ indicate a net charge transfer in the direction of the PDI ligand in the iridium systems. The values of $\Delta_{CT}$ = 7 me$^-$ and 0 me$^-$ for the iridium methyl thiolato and rhodium methoxido complex ($\Delta_{CT}$ = 0 me$^-$), on the other hand, signal a balanced charged transfer between the PDI and M-SMe,OMe fragments. Finally, the positive values for $\Delta_{CT}$ for the residual chlorido and cationic complexes revealed a charge depletion of the PDI ligands with regard to the MX units. This trend of $\Delta_{CT}$ was also reflected in the aggregated QTAIM charges of the PDI ligand, which were most strongly negative in the iridium nitrosyl system (−627 me$^-$) and even increased to positive values for the cationic systems, i.e., +82 and +302 me$^-$ for the rhodium THF and iridium carbonyl complexes. It seems obvious to correlate strongly negative values for $\Delta_{CT}$ and the aggregated PDI ligand charge with the non-innocence of the PDI ligand; accordingly, clear-cut examples were the amido and nitrosyl complexes.

**Table 3.** Binding mode/bond analysis of M-O/S-bonds in (PDI)M-X-R complexes (M = Rh, Ir; X = O, S; R = H, Me, Ph).

| Property/Bond/Complex | | Rh-OMe | Ir-OH | Ir-OMe | Ir-SH | Ir-SMe | Ir-SPh | Ir-NH$_2$ |
|---|---|---|---|---|---|---|---|---|
| **Full Systems** | | | | | | | | |
| Bond distance (M-X) [Å] | exp. [a] | 1.942 | 2.027 | 1.968 | 2.269 | 2.241 | 2.251 | 1.926 |
| | DFT [b] | 1.949 | 1.964 | 1.954 | 2.262 | 2.243 | 2.252 | 1.929 |
| Bond angle (M-X-R) [°] | exp. [a] | 134.01 | n/a [c] | 130.37 | n/a [c] | 118.76 | 122.38 | |
| | DFT [b] | 130.39 | 111.16 | 130.27 | 103.54 | 118.82 | 124.03 | |
| Bond Order Analysis | Wiberg BO | 0.7179 | 0.8547 | 0.7679 | 1.0574 | 1.1407 | 1.0702 | |
| | Wiberg/ Löwdin BO | 0.967 | 1.0615 | 0.9953 | 1.3923 | 1.4068 | 1.3636 | |
| | Fuzzy BO | 1.442 | 1.4573 | 1.4115 | 1.5550 | 1.5938 | 1.6791 | |
| QTAIM Analysis | ρ at BCP(M-X) [au] | 0.1228 | 0.1358 | 0.1364 | 0.1178 | 0.1225 | 0.1191 | |
| | $\nabla^2\rho$ BCP(M-X) [au] | 0.5853 | 0.5573 | 0.5866 | 0.1878 | 0.1856 | 0.1760 | |
| | ellipticity at BCP(M-X) | 0.1471 | 0.2055 | 0.1819 | 0.3345 | 0.1961 | 0.1551 | |
| Local Force Constant k(M-X) [mdyn/Å] [d] | | 2.543 | 2.999 | 3.000 | 2.083 | 2.187 | 2.015 | |
| Bond Dissociation Energy CCSD(T)/def2-TZVPP [kcal/mol] [e] | | 69.28 | 87.90 | 76.36 | 79.08 | 74.36 | 71.03 | |
| **small model complexes** | | | | | | | | |
| C$_s$-symmetry: Wiberg Bond Order a′ σ (and π d$_{xy}$) a″ π d$_{xz}$ C$_{2v}$-symmetry: Wiberg Bond Order a$_1$ σ b$_1$ π d$_{xy}$ b$_2$ π d$_{xz}$ | | 0.729 | 0.872 | 0.819 | 1.098 | 1.165 | 1.106 | 0.975 |
| | | 0.50 | 0.60 | 0.56 | 0.78 | 0.82 | 0.80 | 0.62 |
| | | 0.24 | 0.26 | 0.26 | 0.32 | 0.34 | 0.32 | 0.36 |
| | | | | | | | | 0.56 |
| | | | | | | | | 0.06 |
| | | | | | | | | 0.36 |

[a] det. by X-ray crystallography; [b] geometry-optimized (PBE-DB3J/def2-TZVP/def2-ECP); [c] hydrogen position could not be determined; [d] local force constant. [e] For comparison: BDE((PDI)Rh-Cl): 87.77; BDE((PDI)Ir-Cl): 91.15; BDE((PDI)Ir-Me): 55.64; all values in kcal/mol.

### 4.2.3. Oxidation State Analysis

In order to further judge this assumption, we performed an analysis of the oxidation state of the metal centers and ligands using the LOBA, EOS and OSLO methodologies. For some of the compounds described herein, we previously employed a LOBA for the metal centers, which was complemented by an oxidation state assignment based on experimental XPS and XAS data (Table 2). For our study, we employed the PDI ligand; metal centers and the residual fourth ligand X were defined as fragments. The results are compiled in Table 2. For the OSLO method, the branching option in the QChem program package was employed for borderline cases. For the EOS analysis, different population schemes, e.g., Mulliken and TFVC, were tested, which occasionally provided different results, as denoted in Table 2.

We color-coded Table 2 to indicate the oxidation state of the metal centers and the reduction state of the PDI ligand. A green color indicates an Rh(I) and Ir(I) oxidation state and innocent neutral PDI ligands. This correlates well with the results of the ALMO-EDA and is the case for all complexes with a positive value for $\Delta_{CT}$, i.e., for all systems with a net charge flow from the PDI ligand to the MX units. For the iridium chlorido complex, this assignment agrees with the results of the XPS measurements. Note that the methyl thiolato complex and rhodium methoxido complexes PDI-Rh-OMe belong to this group.

The Ir methoxido compound, PDI-Ir-OMe, on the other hand, is part of the borderline cases, which are color-coded in yellow. The latter also includes the methyl and amido iridium complex PDI-Ir-Me,NH$_2$, for which all negative values were calculated for $\Delta_{CT}$. Both the OSLO and EOS methods (with the QTAIM scheme) indicated iridium(I) oxidation states and neutral PDI ligands for these systems. For the amido complex, this assignment agrees with our findings based on the experimental XAS and XPS data. As previously noted by Head-Gordon et al., the LOBA method seemed to be less reliable for the delocalized electronic structures of the PDI systems and provided oxidation states of +III for the methoxido and amido compound [74]. It deserves special mention that a doubly reduced PDI$^{2-}$ ligand was derived by the OSLO method, which required a cationic methyl group to form a neutral complex ([(PDI$^{2-}$)(Ir$^{+1}$)(Me$^{1+}$)]). Furthermore, different oxidation states were obtained for the amido system, when the EOS (TFVC) scheme was employed. In this case, the complex was described with an Ir(III) center, and negatively charged NH$_2{}^-$ and PDI$^{2-}$ ligands, i.e., an [(PDI$^{2-}$)(Ir$^{+3}$)(NH$_2{}^{1-}$)] electronic structure. This description comes close to the results of the ALMO-EDA, which displayed the largest amount of charge transfer to the PDI ligand (CT(IrNH$_2 \rightarrow$ PDI) = 180 me$^-$) and the overall strongest interaction energy E$_{CT}$(IrNH$_2 \rightarrow$ PDI) = 169 kJ/mol, as well as the second largest aggregated QTAIM charge of the PDI ligand (627 me$^-$). Overall, it should be emphasized that the concept of oxidation states definitely reaches its limit for these highly delocalized systems.

The final group highlighted in orange in Table 2 consists of the nitrosyl and nitrido complexes (X = NO and N), The OSLO and EOS analysis clearly revealed reduced non-innocent PDI ligands for all these systems. With the EOS method, doubly reduced PDI$^{2-}$ ligands and Ir(I) and Ir(III) metal centers were derived for the nitrosyl and nitrido complexes, which is in agreement with previous assignments based on XPS and XAS measurements. For the NO and nitrido ligands, a positive NO$^+$ and negative N$^-$ charge was calculated, which combine to form (PDI$^{2-}$)(NO$^{1+}$)Ir$^{(1+)}$ and (PDI$^{2-}$)(N$^{1-}$)Ir$^{(3+)}$. This is in contrast to the results of the OSLO analysis, which revealed a negative nitrosyl group (NO$^-$) and a quadruply reduced PDI$^{-4}$ ligand, i.e., (PDI$^{4-}$)(NO$^{1-}$)Ir$^{(5+)}$, in contradiction to the XPS data. Overall, these results are consistent with the results of the ALMO-EDA and the total QTAIM charges for the PDI ligand, which showed a large accumulation of charge on the latter.

### 4.2.4. Bonding Metrics and Local Mode Vibrational Analysis

Furthermore, we considered Wieghardt's diagnostic parameter $\Delta_{geo}$ of the PDI ligand, which allows for assigning the reduction state of the PDI ligand, ranging from PDI to PDI$^{-4}$, based on the value of $\Delta_{geo}$. As the rC$_{py}$-C$_{im}$ bond length is expected to correlate with the bond order, we also looked into the local vibrational force constant of this C-C bond by aid of the LMODEA tool (Table 2) [81–83].

The Wieghardt et al. method is based on crystallographic data. While this geometric approach has great applicability, limitations arise when dealing with poor or distorted crystal structures. We therefore also calculated the $\Delta_{geo}$ parameter for DFT-optimized geometries (Tables 1 and 2). In agreement with the results of our analysis presented in Sections 4.2.2 and 4.2.3, small $\Delta_{geo}$ parameters in the range of 0.037–0.067 Å were observed for the doubly reduced PDI ligands in the (PDI)IrX complexes for X = NO, N (orange color in Table 2). Consistently, these compounds also displayed short rC$_{py}$C$_{im}$ bond distances, which reached the range of C-C double bonds in the nitrosyl complex (rC$_{py}$C$_{im}$ = 1.377 Å). This was also reflected in the high values for the corresponding local vibrational force constants of these complexes, which lies above 6 mdyn/Å for the nitrosyl compound.

The local vibrational force constant of the rC$_{py}$C$_{im}$ bond for the members of the second group with innocent PDI ligands and strong back-donation (M = Ir, X = NH$_2$, OMe, Me, color encoded in yellow) was also high, indicating a clear double-bond character. Unexpectedly, the experimental value of 0.036 Å for the $\Delta_{geo}$ parameter in the iridium amido complex reached the one for the nitrosyl complex. This might be explained by the rather poor X-ray crystal structure of the amido complex, which is supported by a comparison of the $\Delta_{geo}$ parameters of 0.037 Å (X = NO) and 0.063 Å (X = NH$_2$) for the DFT-

optimized geometries of the full system. Combining the results of all the diagnostic tools, it becomes apparent that the amido system is borderline. The PDI ligand experienced a very strong charge transfer from the IrNH$_2$ unit, but may yet be considered neutral/innocent. The iridium methoxido and methyl complexes displayed larger values of 0.083 and 0.091 Å, consistent with a sizable but reduced charge transfer to the PDI ligand. Overall, these data are therefore consistent with the assignment to the yellow-coded entries in Table 2 for complexes with PDI ligands on the brink of non-innocence.

The last category encoded in green in Table 2 consists of complexes with innocent PDI ligands. This is most clear-cut for the cationic complexes, displaying $\Delta_{geo}$ parameters of >0.1 Å for the unambiguously innocent PDI ligands, which was most pronounced for the (PDI)Ir-CO complex with the CO π-acceptor ligand ($\Delta_{geo}$ = 0.155 Å). This cationic carbonyl complex also displayed the longest C$_{py}$-C$_{im}$ bond distance (rC$_{py}$C$_{im}$ = 1.481 Å) and the smallest local vibrational constant (4.59 mdyn/Å) of all the compounds. On the other end of this group, complexes with SMe and OMe π-donor ligands were found. For the iridium methyl thiolato complex **14**, the values of 0.078 Å and 1.439 Å for $\Delta_{geo}$ and rC$_{py}$C$_{im}$ as well as the corresponding large local vibrational constants of 5.426 and 4.59 mdyn/Å, respectively, reached or surpassed those of the yellow-encoded entries in Table 2. This clearly signals substantial π back-donation to the PDI ligand, which is only partially reflected in the results of the ALMO-EDA (vide supra).

In conclusion, a nearly seamless transition can be observed from the cationic iridium carbonyl complex, characterized by minimal π back-bonding and an innocent, neutral PDI ligand, to the nitrosyl complex featuring a non-innocent, doubly reduced PDI$^{2-}$ ligand.

### 4.2.5. Characterization of the M-X Bond

In our initial paper on PDI complexes over two decades ago, we emphasized the remarkably short Ir-O bond length of 1.949(4) Å in a methoxido iridium complex [26]. At that time, this bond distance was the shortest among d$^8$-configured square-planar iridium complexes containing hydroxido, alkoxido, or aryloxido ligands. We hypothesized that this observation could be attributed, at least partially, to the weaker trans influence of the pyridine unit in the PDI complexes compared to related L$_3$Ir-OR (R = H, alkyl, aryl) compounds, which predominantly featured CO or phosphine ligands in the trans position.

One intriguing characteristic of the aforementioned iridium and methoxido PDI complexes is their remarkable thermal stability. When heated in C$_6$D$_6$ for several days, these compounds exhibited minimal decomposition, with primarily unreacted starting material remaining. This starkly contrasted the facile β-hydrogen elimination process observed in square-planar rhodium and iridium complexes in the presence of phosphine donors, as reported in the literature [6,13]. Relevant theoretical and kinetic studies were also reported for nickel and palladium methoxido PCP–pincer systems [85].

DFT calculations showed that the β-hydrogen elimination leading to formaldehyde and the corresponding hydrido complex is highly thermodynamically unfavorable by more than 30 kcal/mol [26]. This exceptional stability of the M-O bond was attributed to favorable π-donation of the O-donor rather than a destabilizing 4-electron 2-orbital destabilization of the occupied d$_{xz}$ orbital and the p-orbital of the oxygen atom, i.e., the well-known dπ-pπ repulsion between electron-rich late-transition-metal centers and π-donor ligands [7,10,12,86]. This is due to an empty PDI π* orbital and results in a rotational barrier for the rotation around the M-O bond. In our previous DFT calculations, we estimated an activation barrier of approximately 12 kcal/mol for this rotation. Despite our attempts, (cf below) we have not yet been able to experimentally confirm this hypothesis.

Hence, when we started to look into the chemistry of the corresponding thiolato complexes, we revisited the analysis of the dπ-pπ interaction including more recent theoretical diagnostics, e.g., NBO [67] and QTAIM [64] analysis. Furthermore, as described below, we were able to present experimental evidence supporting the existence of a rotational barrier in the methyl thiolato complex.

We will begin by comparing the Rh-O, Ir-O and Ir-S bond lengths with values reported in the literature. Regarding the M-O bond distances, it is worth noting that the value of 2.027 Å observed in the iridium hydroxido complex **7** falls outside the narrow range of 1.94–1.97 Å (Table 1). We attribute this deviation to certain problems encountered during the X-ray crystal structure determination, as detailed in the Supplementary Materials. This conclusion is supported by the shorter bond distance of 1.965 Å observed in the previously studied analogous rhodium hydroxido complex **6** and the value of 1.964 Å obtained from the DFT-optimized geometry. By comparing our PDI complexes to the average of 2.05 ± 0.05 Å reported for Rh/Ir-O bonds in the current Cambridge Structural Database (CSD) for square-planar $L_3$M-OR complexes (M = Rh, Ir, R = H, alkyl, aryl), it becomes clear that the M-OR bonds in our PDI complexes continue to exhibit exceptionally short M-O distances within this molecular class. For further information on the CSD search, please refer to the Supplementary Materials. It is worth mentioning that the Schneider group observed another instance of a short M-O bond length, measuring 1.935 Å, in a (PNP)Ir(III) hydroxido complex [87].

The Ir-S bond distances in our PDI Ir sulfido complexes, as described herein, range from 2.241 Å to 2.269 Å. These values are extremely well-reproduced in the DFT-optimized geometries (Table 3). The analysis of rhodium and iridium sulfur bond distances in square-planar complexes reported in the CSD revealed substantially longer bonds averaging at 2.37 ± 0.04 Å (details see Supplementary Materials). There is only one example reported by Braun et al. for a (POP)Rh-SH pincer complex, which displayed an Rh-S bond distance of 2.286 Å [37], coming close to the very short values observed in the compounds described herein.

Taking into consideration the moderate to weak trans influence of the pyridine ligand [88,89], the remarkably short Ir-S bond distances in the PDI complexes suggest the presence of strong Ir-S bonds with a multiple-ligand bond character. This was confirmed by calculating the Wiberg and Fuzzy bond orders (BOs), which were found to be greater than 1 in the $C_s$-symmetrical model complexes listed in Table 3. To further analyze the bond orders, the AOMIX program package was utilized to decompose the Wiberg/Mayer bond orders into their symmetry components for the corresponding small model complexes [79].

In $C_s$-symmetry, the contributions of orbitals with a″ symmetry can be clearly assigned to Ir-S π-bonding interactions involving the iridium $d_{xz}$ and sulfur $p_z$ orbitals. On the other hand, MOs with a′ symmetry are related to a combination of σ- and π-bonding involving the $d_{x^2-y^2}/d_{z^2}$ and $d_{xy}$ orbitals, respectively (using $D_{4h}$-symmetry notation). Examining Table 3, it is evident that π-bonding plays a significant role, contributing values exceeding 0.3 to the Wiberg/Mayer bond order, which corresponds to approximately 40%. For comparison, a similar analysis was performed for the previously isolated $C_{2v}$-symmetrical PDI iridium amido complex [22], allowing for a further decomposition of contributions into their σ- and π-components. This calculation revealed a comparable contribution of the MOs with $d_{xz}$ participation to the bond order, while those involving $d_{xy}$ orbitals exhibited a substantial reduction. Additionally, while smaller Mayer bond orders were calculated for the M-O bonds, partial π-bonding was also identified through the symmetry decomposition in the metal methoxido and hydroxido complexes (Table 3).

The π-character of the M-O,S bonds was confirmed by the results of the QTAIM (Quantum Theory of Atoms in Molecules) analysis [64], which identified bond critical points (BCP(3, −1)) for all M-O and M-S bonds. The corresponding bond ellipticities deviated significantly from zero, indicating non-rotational symmetry of the electron density at the BCP and thus confirming the presence of π-bonding [64]. The electron densities at the bond critical points of the M-O and M-S bonds were substantial (ranging from 0.118 to 0.136 e/Å$^3$). Combined with the positive values for the Laplacian $\nabla\rho^2$, this suggests an ionic character for the M-X bonds [90], which aligns with the electronegativity differences between the metal centers and the chalcogenides (EN (Pauling scale): Rh: 2.28; Ir: 2.20; O: 3.44; S: 2.58).

We have also calculated the bond dissociation energies at the PNO-CCSD(T)/def2-TZVPP//PBE-D3BJ/TZVP level of theory. In this regard, a previous work by Ess et al. focusing on the binding of late transition metals (Ru, Rh, Ir, Pt) to heteroatoms deserves special mention [8]. They employed DFT and canonical CCSD(T) calculations with a small basis set (6–31 G(d,p)) due to computational and methodological limitations at that time. However, their study successfully reproduced the experimentally established correlation between M-X and H-X bond energies [91]. The values of the bond dissociation energies for the (PDI)M-O and (PDI)M-S ranged from ca. 70–88 kcal/mol, suggesting rather strong bonds (Table 3). As expected, the M-O bond in the methoxido compounds was ca. 7 kcal/mol higher in the iridium complex compared to the 4d rhodium congener, which we attribute to relativistic effects. Furthermore, in agreement with previously established correlations for H-X with M-X bond dissociation enthalpies [91] (e.g., BDE(H-OH) = 118.9 kcal/mol, BDE(H-SH) = 91.29 kcal/mol), weaker bonds were observed in the thiolato complexes (Ir-OH: 87.9 vs. Ir-SH: 79.08 kcal/mol).

Additional support for the $\pi$-bonding nature of the M-O and M-S bonds was provided by NBO (Natural Bond Orbital) analysis of the methoxido and methyl thiolato iridium model compounds. This analysis revealed clear $sp^2$ hybridization of the oxygen and sulfur atoms. In the case of the methyl thiolato complex, four doubly occupied lone pairs (LPs) were assigned to the $d_{xy}$, $d_{yz}$, $d_z{}^2$, and $d_{xz}$ orbitals of the $d^8$-configured iridium(I) center (see Supplementary Materials). However, the situation is more complex for the methoxido compound. The leading NBO structure involves only three LPs for the Ir center, representing the $d_{xy}$, $d_{yz}$, and $d_z{}^2$ orbitals. The $d_{xz}$ orbital is engaged in a $\pi$-bonding interaction with the pyridine nitrogen orbital, with approximately equal contributions from the nitrogen and iridium atoms (Figure 3).

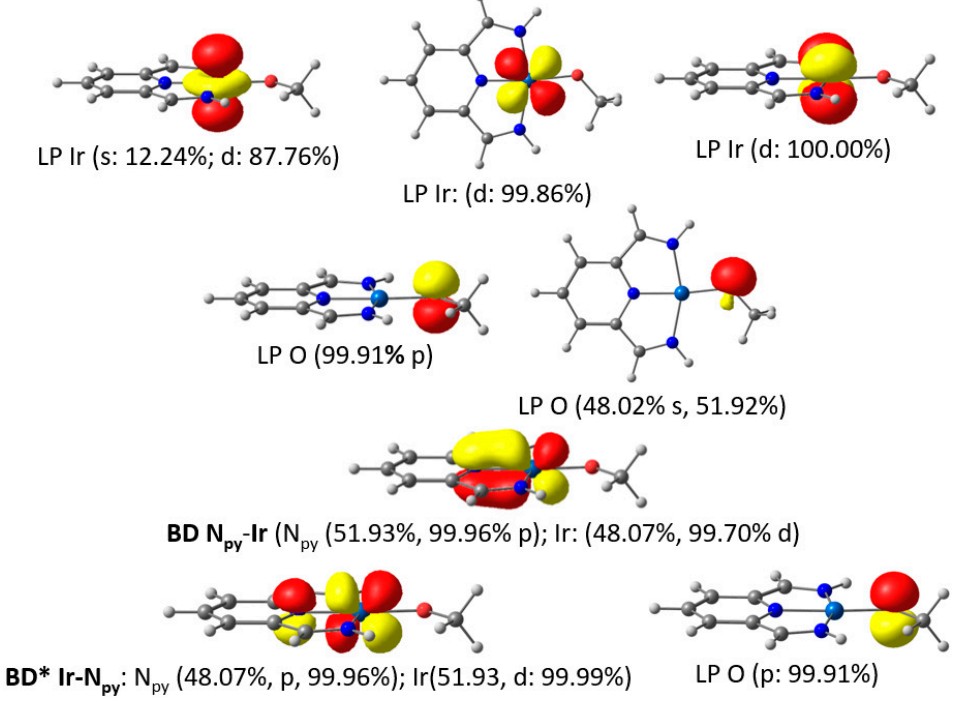

LP Ir (s: 12.24%; d: 87.76%)

LP Ir: (d: 99.86%)

LP Ir (d: 100.00%)

LP O (99.91% p)

LP O (48.02% s, 51.92%)

**BD $N_{py}$-Ir** ($N_{py}$ (51.93%, 99.96% p); Ir: (48.07%, 99.70% d)

**BD* Ir-$N_{py}$**: $N_{py}$ (48.07%, p, 99.96%); Ir(51.93, d: 99.99%)

LP O (p: 99.91%)

**Figure 3.** NBOs related to the Ir-O bonding in the iridium methoxido model complex with population numbers.

The NBO analysis further revealed an Ir-O $\sigma$-bond and two additional lone pairs of the oxygen atom in the methoxido complex. One of these lone pairs corresponded to the $sp^2$ orbital located in the square plane, while the other represented a pure $p_z$ orbital. The latter exhibited a strong second-order perturbation interaction of 31.3 kcal/mol and

contributed to the antibonding combination of the NBO associated with the aforementioned $N_{pyridine}$-Ir($d_{xz}$) $\pi$-bond (Figure 3).

### 4.3. Experimental Evidence for M-X $\pi$-Bonding–M-X-R Rotational Barrier

Due to the partial $\pi$-bond character of the M-XR (X = O, S) bond, a rotational barrier for the M-X bond could be envisaged. In the solid state, the alkoxido and hydroxido as well as the thiolato compounds display an idealized $C_s$-symmetry with bent $sp^2$-hybri-dized M-O,S-R units located in the square plane (Table 1 and Figure 1). In solution, however, the vT $^1$H and $^{13}$C NMR spectra are consistent with a time-averaged $C_{2v}$-symmetry of the complexes for all but the methyl thiolato compound **14**. Of particular diagnostic use in this regard are the protons of the pyridine ring, which exhibit a triplet for the para proton and a doublet for the (time-averaged) homotopic (or enantiotopic) meta protons with an integration ratio of 1:2. For the methyl thiolato complex **14**, we observed temperature-dependent $^1$H spectra, with a splitting of the meta pyridine protons into two separate diastereotopic resonances (doublets) at <263 K, consistent with the idealized $C_s$-symmetrical structure found in the solid state and in the DFT-optimized geometry (Figure 1 and Supplementary Materials). Upon warming, these signals broadened, coalesced at 318 K in the 400 MHz spectrum and turned into a doublet at elevated temperatures (Figure 4 left and Supplementary Materials).

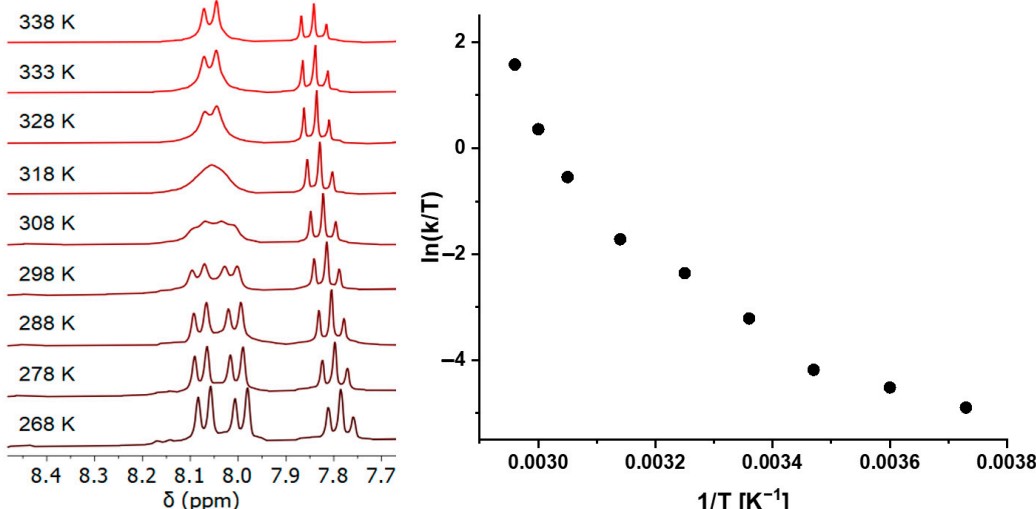

**Figure 4.** Pyridine section of the vT $^1$H NMR spectra of the iridium methyl thiolato complex **14** (**left**) and Eyring plot for the rate constants derived from line shape analysis of the $^1$H NMR data (**right**).

This reversible dynamic process was modeled by line shape analysis (LA) of the vT $^1$H NMR spectra in toluene in the temperature range of 268–338 K (details see Supplementary Materials). It deserves special mention that the derived barrier at the coalescence temperature of $\Delta G^{\#}$ = 15.7 kcal/mol was consistent with the estimate by the Gutowski–Holm equation, which gave $\Delta G_{318}^{\#}$ = 16.2 kcal/mol. With the fitted rate constants of the LA, we attempted to obtain the activation parameters from an Eyring plot (Figure 4 right).

Much to our surprise, rather than a linear dependence of ln(k/T) on the reciprocal temperature, we clearly obtained a curved line, which is an indication of (at least) two independent processes with different activation parameters [92].

Besides rotation about the M-X bond shown below in Scheme 4 (item a), other processes (items b–e) were considered to explain the observed time-averaged spectra.

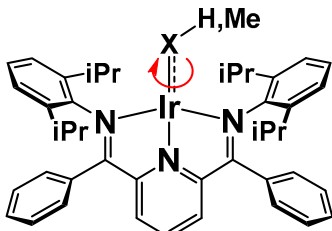

**Scheme 4.** a. Rotation about the Ir-X bond.

This further includes:

b. R = H, Me, Ph: ionization and recombination of the ions (PDI)M-XR $\leftrightarrows$ (PDI)M$^+$ + XR$^-$

c. R = H: sequence of $\alpha$-H elimination (IrOH, IrSH) and the microscopic reverse 1,2-H shift (less likely for R = Me and Ph) (PDI)M-X-H $\leftrightarrows$ (PDI)M(H) = X

d. "windshield wiper" process with a $C_{2v}$-symmetrical transition state and sp-hybridized oxygen or sulfur atoms

e. R = Me: $\beta$-H elimination, (rotation about of the form/thioaldehyde unit) and reinsertion (PDI)M-X-H $\leftrightarrows$ (PDI)M(H)($\eta^2$-CH$_2$X) ($\leftrightarrows$ rotation about (PDI)M(H)-($\eta^2$CH$_2$X) $\leftrightarrows$ (PDI)M-X-H)

The values for the rotational barriers about the M-X bonds (item a) were estimated from the corresponding transition states and are compiled in Table 4 for the iridium methoxido and thiolato complexes.

**Table 4.** Rotational barriers calculated at the DFT and LNO-CCSD(T) levels according to Scheme 4 (item a) and bond orders and distances in the ground and transition states.

| Method/Complex | TS(IrOMe) | TS(IrSH) | TS(IrSMe) | TS(IrSMe)THF |
|---|---|---|---|---|
| $\Delta E^{\#}$ [kcal/mol] DFT(PW6B95-D3BJ(def2-TZVP))/ PBE-D3BJ(def2-TZVP) | 9.06 | 13.70 | 15.43 | 15.87 |
| $\Delta E^{\#}$ [kcal/mol] LNO-CCSD(T)(def2-TZVPP)/ PBE-D3B-J(def2-TZVP) | 8.76 | 14.29 | 15.02 | 17.22 |
| $\Delta G^{\#}_{298}$ [kcal/mol] LNO-CCSD(T)(def2-TZVPP)/ PBE-D3B-J(def2-TZVP) | 8.15 | 16.08 | 16.68 | 18.07 |
| Wiberg Bond Order(GS)/ distance (Ir-X) [Å] | 0.768/1.954 | 1.057/2.264 | 1.141/2.243 | 1.141/2.243 |
| Wiberg Bond Order(TS)/ distance (Ir-X) [Å] | 0.630/1.980 | 0.877/2.379 | 0.884/2.343 | 0.930/2.325 |

First of all, the good agreement between the DFT and LNO-CCSD(T) calculations deserves special mention. There was only a very small rotational barrier of ca. 9 kcal/mol for the iridium methoxido complex **9**. The larger activation energies of ca. 14 (SH) and 15 kcal/mol (SMe) were derived for the thiolato complexes **14** and **16**. It is assumed that in contrast to the methyl thiolato complex **14,** the smaller rotational barrier of 14 kcal/mol for the terminal hydrosulfido ligand in **16** may be just slightly too small to manifest itself in the vT $^1$H NMR spectra. The transition state for the methyl thiolato compound **14** depicted in Figure 5 (left) clearly shows that the S-methyl group is no longer located in the square plane, but rather oriented perpendicular to it. This leads to an idealized $C_s$-symmetry, which gives rise to enantiotopic meta-pyridine protons in the $^1$H NMR spectrum. Inspection of Table 4 reveals that the Ir-X bonds are elongated in the transition state, which is accompanied by a sizable reduction of the corresponding bond orders. We anticipate that this is a consequence of the missing d$\pi$-p$\pi$ interaction in the transition state.

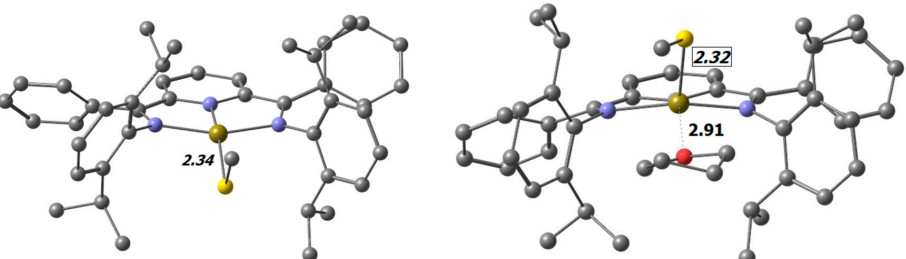

**Figure 5.** Transition states for rotation about the Ir-S bond. **Left**: without the contribution of the solvent; **Right**: with the contribution of THF. Selected distances are shown in Å. (C: ●, N: ●, Ir: ●, O: ●, S: ●).

In the search for another process contributing to the non-linear Eyring plot, we identified an additional five-coordinate transition state, which involved coordination of the THF solvent to the iridium center in the transition state (Figure 5 right).

The calculated activation energy of 15.87 kcal/mol (DFT) is only slightly larger than the barrier in the four-coordinate complex (15.43 kcal/mol). Although the energy difference between these two transition states comes out slightly larger in the LNO-CCSD(T) calculation, we anticipate that this process is certainly competitive and might hence serve as a good explanation for the curved Eyring plot.

b.    Ionization and recombination

For the ionization process, we calculated the reaction energy $\Delta E$ of the full iridium methyl thiolato complex **14** in THF according to Equation (1) at the DFT PBE-D3BJ(def2-QZVPPD)/COSMO ($\varepsilon = 7.6$) level.

$$(PDI)Ir\text{-}SMe + THF \leftrightarrows (PDI)Ir\text{-}THF^+ + SMe^-, \quad \Delta E_R = +45.5 \text{ kcal/mol} \qquad (1)$$

The COSMO dielectric continuum solvation model might not fully and adequately describe the true picture; therefore, the calculated value of +45.5 kcal/mol has to be considered an upper limit. It should be noted, however, that ionization is uphill by 28.7 kcal/mol even for solvation with water ($\varepsilon = 80$). Therefore, we anticipate that the reaction energy is too endothermic to compete with other processes. This is also in agreement with our previous molecular conductivity measurements of a PDI iridium complex bearing a significantly better ionizable triflato ligand, which is only partially ionized in THF solution [26].

c.    $\alpha$-H Elimination and the microscopic reverse step

The $\alpha$-H elimination and its reverse process, the [1,2] H-shift, is well-established for alkyl ligands [93], but for oxo and sulfido ligands this reaction is less common [86,94]. We calculated the thermodynamics for the formation of the hydrido, oxo/sulfido complex by $\alpha$-H elimination in the hydroxido and hydrosulfido complexes. The DFT-optimized geometries of the $\alpha$-H elimination products revealed slightly distorted square-pyramidal structures, which are shown in Figure 6. It deserves special mention that the Ir-S bond distance was only marginally shortened in the product (2.22 vs. 2.24 Å), while the Ir-O bond length dropped by 0.14 Å.

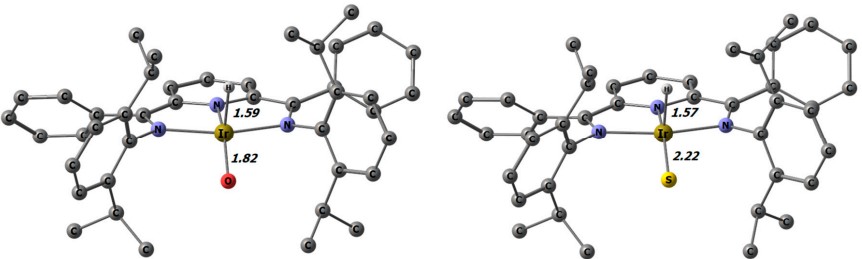

**Figure 6.** $\alpha$-H elimination products of **7** and **16** with selected distances in Å.

Both products shown in Figure 6 are substantially energetically uphill from their corresponding hydroxido and hydrosulfido educts. The values of +50.9 kcal/mol for the oxido and +31.4 kcal/mol for the sulfido complex calculated at the DFT level with the PW6B95-D3BJ functional (def2-TZVP basis) clearly rule out that the α-H elimination step plays a role in the observed dynamic behavior.

d.    "Windshield" wiper process

The "windshield wiper" process refers to the in-plane movement of the O,S-R bond group (R = H, Me, Ph) from one molecule side to the other. This process was modeled for the unsubstituted methyl thiolato complex; the corresponding Walsh diagram is presented in Figure 7.

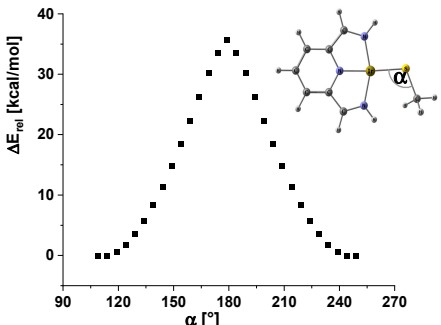

**Figure 7.** Walsh diagram for the windshield wiper process in the methyl thiolato iridium model complex.

The windshield wiper process exhibited an activation energy of approximately 40 kcal/mol. The transition state adopted a $C_{2v}$-symmetrical structure, in which the hydrogen atoms of the methyl group were disregarded. For the complete system, an even higher barrier is expected; consequently, it is concluded that this process is not involved in the observed dynamic behavior.

e.    β-H Elimination and reinsertion

For the methyl-substituted methoxido and methyl thiolato complexes, ß-hydrogen elimination leading to $C_s$-symmetrical hydrido $\eta^2$-formaldehyde and thioformaldehyde complexes can be envisaged. This process was studied by DFT and LNO-CCSD(T) calculations; the geometry-optimized structures of the transition states are depicted in Figure 8.

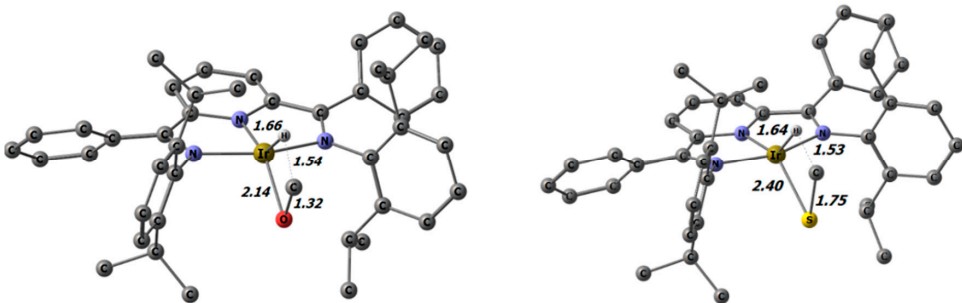

**Figure 8.** Transition states for ß-hydrogen elimination in the methoxido and methyl thiolato iridium complexes **9** and **14** with selected distances in Å.

The activation energies calculated at the DFT PW6B95-D3BJ/def2-TZVP level for the iridium methoxido and methyl thiolato complexes amounted to 16.7 and 23.4 kcal/mol, respectively. These barriers are approximately 7 kcal/mol higher than those for the rotation about the M-O and M-S bonds. Therefore, the β-hydrogen elimination processes are not considered to be competitive with regard to the observed dynamic behavior. However, it has to be emphasized that these β-hydrogen elimination barriers are quite low. Nevertheless,

further elimination of the thio-/aldehyde to yield the hydrido complex was not observed. This contrasts with the results of other late-transition-metal alkoxide complexes reported in the literature. Frequently, these compounds exhibit relatively low thermal stability, which is due to β-hydrogen elimination and the consecutive elimination of the aldehyde or ketone [9,13,85,95,96]. For the PDI complexes, this difference can be rationalized by the fact that extrusion of the aldehyde/thioaldehyde is highly thermodynamically unfavorable (Figure 9). At the LNO-CCSD(T) level, this step is 25.58 (X = O) and 50.05 (X = S) kcal/mol uphill (Table 5).

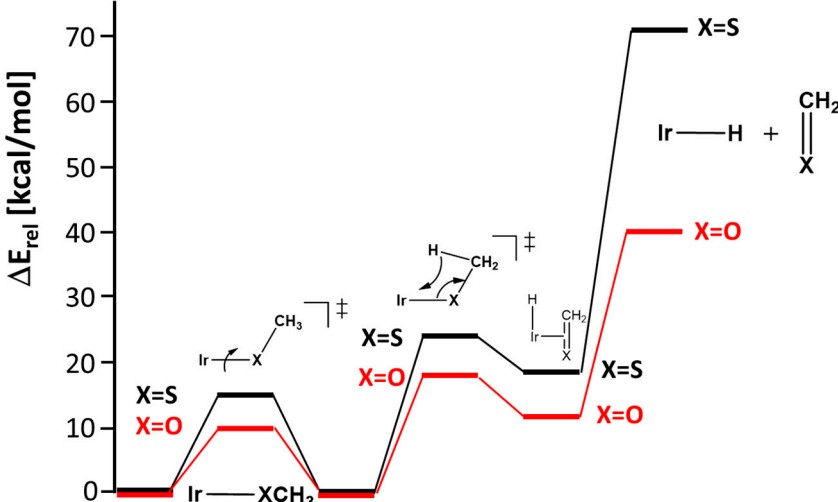

**Figure 9.** Calculated energy profile at the LNO-CCSD(T)/def2-TZVPP//DFT(PBE-D3B)/def2-TZVP level for the rotation about the Ir-X bond (X = O,S) in complexes **9** and **14,** and ß-hydrogen elimination with consecutive extrusion of thio-/formaldehyde.

**Table 5.** Energy levels for ß-hydrogen elimination and thio-/formaldehyde extrusion in the iridium methoxido and methyl thiolato complexes **9** and **14**.

| STEP/Method | $\Delta E^{\#}_{rel}$ [kcal/mol] DFT(PW6B95-D3BJ (def2-TZVP)) | | $\Delta E^{\#}$ [kcal/mol] LNO-CCSD(T) (def2-TZVPP) | |
|---|---|---|---|---|
| Complex | X = O | X = S | X = O | X = S |
| TS(ß-H elimination) | 17.91 | 24.21 | 18.27 | 22.97 |
| intermediate | 11.82 | 18.48 | 11.71 | 16.28 |
| product | 39.94 | 70.80 | 37.29 | 66.33 |
| product–intermed. | 28.12 | 52.32 | 25.58 | 50.05 |

The reverse association process of formaldehyde is essentially barrierless, as can be recognized from the inspection of the linear transit for this process shown in Figure 10.

Finally, we investigated the isomerization pathway from the methoxido to the hydroxymethylene complex according to Equation (2).

$$(\text{PDI})\text{Ir-OCH}_3 \leftrightarrows (\text{PDI})\text{Ir-CH}_2\text{OH} \tag{2}$$

This type of transformation was reported by Wayland et al. for a rhodium porphyrin system and was found to be energetically in favor of the hydroxymethylene complex by ca. 6 kcal/mol [97]. We studied this reaction by DFT and LNO-CCSD(T) calculations, which are summarized in Figure 11.

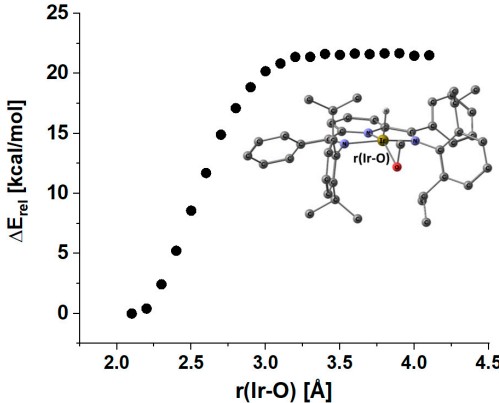

**Figure 10.** Energy profile for extrusion or association of formaldehyde in the iridium hydrido $\eta^2$-formaldehyde complex at the r2scan-3c DFT level (def2-mTZVPP basis; ECP: def2-ECP).

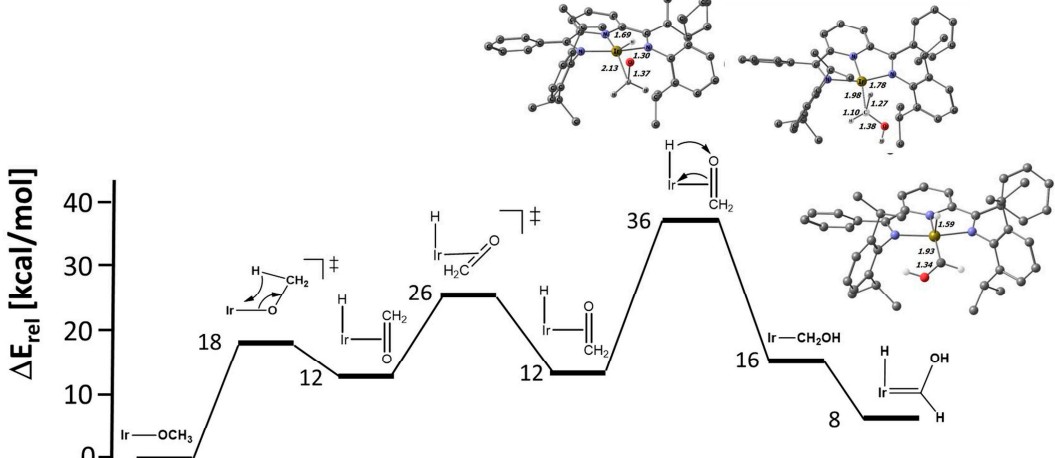

**Figure 11.** Energy profile at the LNO-CCSD(T)/def2-TZVPP//DFT(PBE-D3BJ/def2-TZVP) level for the transformation of the iridium methoxido complex **9** to the hydroxymethylene and hydroxycarbene congeners. Selected distances in Å are shown.

In contrast to the porphyrin system, the formation of the hydroxymethylene system is significantly uphill for both the Rh (+24.2 kcal/mol) and Ir (18.8 kcal/mol) complexes. We take this as one more hint that the $d\pi$-$p\pi$ interaction in the methoxido complexes is stabilizing. We noted two more interesting features in these calculations. First of all, there is an apparent $\alpha$–H–agostic interaction in the hydroxymethylene complexes, as indicated by the short M$\cdots$H-CHOH interactions of 1.87 (M = Rh) and 1.78 Å (M = Ir) with concomitant elongated C-H bonds of 1.19 and 1.27 Å (see Figure 11 for M = Ir and ESI for M = Rh). These C-H bonds are on the brink of $\alpha$-elimination to the corresponding hydrido hydroxycarbene complexes. For iridium, the carbene compound lies only 7.7 kcal/mol above the methoxido complex, while the rhodium carbene system is significantly less stable (+17.7 kcal/mol). As expected for the multiple-bond character in the hydroxycarbene complexes, the M-C bonds are shortened with respect to the hydroxymethylene complexes (Ir: 0.05 Å, Rh: 0.08 Å) (Figure 11 and ESI).

Finally, this section can be concluded by saying that the (PDI)M-O,S-R complexes (M = Rh, Ir, R = H, Me, Ph) display significant thermal stability, which can be attributed to partially stabilizing $d\pi$-$p\pi$ interactions. Complexes with methoxido and methyl thiolato ligands are particularly noteworthy. Despite their small barriers for ß-hydrogen elimination, they are stabilized by the thermodynamically highly unfavorable extrusion of the thio-/aldehyde in the consecutive step. Finally, it can therefore be concluded that only the rotational process with or without solvent contribution is involved in the dynamic process.

### 5. Conclusions

In this article, we analyzed the bonding situation in square-planar PDI rhodium and iridium complexes with a particular focus on O,S π-donors. The PDI ligand serves as an electron acceptor and assists in reducing the destabilizing electron-2 orbital orbital dπ-pπ interactions between the metal centers and the O,S–heteroatom ligands. Consequently, a partial Ir-O,S multiple-bond character can be derived, which is also reflected in the observed sizable Ir-S rotational barrier and the large calculated homolytic BDEs. We are currently in the final stages of our investigation on the hydrogenation of the hydroxido, methoxido and thiolato complexes and will provide a comprehensive report in the near future.

**Supplementary Materials:** The following supporting information can be downloaded at: https://www.mdpi.com/article/10.3390/chemistry5030133/s1, Figure S1: NMR assignment for the respective protons and carbon atoms; Figure S2: $^1$H NMR spectra of 2 in dibromomethane-d$_2$ between 296 K and 353 K; Figure S3: $^1$H NMR spectra of 14 in toluene-d8 between 268 K and 338 K; Figure S4: ATR-IR spectra of 6 (Rh-OH) and the deutero isotopologue **6D** (RH-OD); Figure S5: Ortep diagram of the molecular structure of **1**. Hydrogen atoms are omitted for clarity, ellipsoids are shown at the 50% probability level; Figure S6: Ortep diagram of the molecular structure of **2**. Hydrogen atoms and solvent molecules are omitted for clarity, ellipsoids are shown at the 50% probability level; Figure S7: Ortep diagram of the molecular structure of **4**. Hydrogen atoms and solvent molecules are omitted for clarity, ellipsoids are shown at the 50% probability level; Figure S8: Ortep diagram of the molecular structure of **6**. Hydrogen atoms with exception of the OH group and solvent molecules are omitted for clarity, ellipsoids are shown at the 50% probability level; Figure S9: Ortep diagram of the molecular structure of **7**. Hydrogen atoms with exception of the OH group and solvent molecules are omitted for clarity, ellipsoids are shown at the 50% probability level; Figure S10: Ortep diagram of the molecular structure of **8**. Hydrogen atoms and solvent molecules are omitted for clarity, ellipsoids are shown at the 50% probability level; Figure S11: Ortep diagram of the molecular structure of **9**. Hydrogen atoms and solvent molecules are omitted for clarity, ellipsoids are shown at the 50% probability level; Figure S12: Ortep diagram of the molecular structure of **10**. Hydrogen atoms, solvent molecules and the counter ion are omitted for clarity, ellipsoids are shown at the 50% probability level; Figure S13: Ortep diagram of the molecular structure of **11**. Hydrogen atoms except of the OH group, solvent molecules and the counter ion are omitted for clarity, ellipsoids are shown at the 50% probability level; Figure S14: Ortep diagram of the molecular structure of **12**. Hydrogen atoms, solvent molecules and the counter ion are omitted for clarity, ellipsoids are shown at the 50% probability level; Figure S15: Ortep diagram of the molecular structure of **13**. Hydrogen atoms, solvent molecules and the counter ion are omitted for clarity, ellipsoids are shown at the 50% probability level; Figure S16: Ortep diagram of the molecular structure of **14**. Hydrogen atoms and solvent molecules are omitted for clarity, ellipsoids are shown at the 50% probability level; Figure S17: Ortep diagram of the molecular structure of **15**. Hydrogen atoms and solvent molecules are omitted for clarity, ellipsoids are shown at the 50% probability level; Figure S18: Ortep diagram of the molecular structure of **16**. Hydrogen atoms with exception of the SH group and solvent molecules are omitted for clarity, ellipsoids are shown at the 50% probability level; Figure S19: Ortep diagram of the molecular structure of **18**. Hydrogen atoms and solvent molecules are omitted for clarity, ellipsoids are shown at the 50% probability level; Table S1: Selected averaged bond distances and angles determined by X-ray crystallography with DFT-optimized values (PBE-D3BJ/def2-TZVP, Rh, Ir, def2-ECP) in parentheses and $^{13}$C NMR data related to the PDI unit; Table S2: Summary of the crystal data and structure refinement of **1**, **2** and **4**; Table S3: Summary of the crystal data and structure refinement for the reported complexes **6**, **7** and **8**; Table S4: Summary of the crystal data and structure refinement for the reported complexes **9**, **10** and **11**; Table S5: Summary of the crystal data and structure refinement for the reported complexes **12**, **13** and **14**; Table S6: Summary of the crystal data and structure refinement for the reported complexes **15**, **16** and **18**; Figure S20: Rh,Ir-O Bond Distances and employed structures by CCSD code; Figure S21: Rh,Ir-S bond distances and employed structures by CCSD code; Figure S22: NBOs of (PDI)Ir-SMe model complex: LPs of the Ir and S atoms. References [26,33–35,48,49,51,62,65,68,69,75–78,81–83,98–108] are cited in the Supplementary Materials.

**Author Contributions:** Conceptualization, P.B.; methodology, P.B. and M.S. (Michel Stephan); validation, M.S. (Michel Stephan), M.V. and P.B.; investigation, P.B., M.S. (Michel Stephan), M.V. and M.S. (Matthias Schreyer); resources, P.B. (University of Hamburg); data curation, M.S. (Michel Stephan) and P.B.; writing—original draft preparation, P.B.; writing—review and editing, P.B., M.S. (Michel Stephan) and M.V. visualization, P.B. and M.S. (Michel Stephan); supervision, P.B.; project administration, P.B.; funding acquisition, P.B. All authors have read and agreed to the published version of the manuscript.

**Funding:** This research received funding by the University of Hamburg.

**Data Availability Statement:** The X-ray crystallographic data were submitted to the Cambridge Crystallographic Database and can be accessed via the deposition numbers: CCD 2282018-2282033.

**Acknowledgments:** We are indebted to Pedro Salvador Sedano for generously sharing the APOST-3D code and for his invaluable assistance in conducting the EOS analysis. We are grateful to QChem for providing a pre-release of QChem rel. 6.1, which allowed us to make full use of the branching option of the OSLO method, and would also like to acknowledge the support and assistance of Abdul Aaldossary. We would like to thank Marc Prosenc for discussions on the NBO analysis and assistance with the X-ray crystal structure refinement of the twinned crystal of complex **9**.

**Conflicts of Interest:** The authors declare no conflict of interest.

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
