# Peer review of "Syntheses, Crystal and Electronic Structures of Rhodium and Iridium Pyridine Di-Imine Complexes with O- and S-Donor Ligands: (Hydroxido, Methoxido and Thiolato)"

_chemistry, doi:10.3390/chemistry5030133_

Round 1
Reviewer 1 Report
In this manuscript the bonding properties in square-planar PDI rhodium and iridium complexes are discussed along with analysis of structural dynamics influencing spectra, as well as the hydrogenation reactions.
My main problem is that the paper is extremely long (30 pages without references). In my opinion, it could easily be split into two papers, one on the structure and one on the structural dynamics. Section five on the hydrogenation reactions should be skipped (see below). The authors fail to give a strong case of using the structural conclusions in the other parts, thus the manuscript falls into parts by itself.
This suggestion is also proven by the fact that the authors failed to present a uniform paper: there are repeating statements, definitions, thoughts. For example, Delta_geo is introduced and discussed in line 328, and later a full section (section 3.5) is devoted to this.
I have serious problems with the tables. In general, there are too many different quantities provided in the tables and these are not well described in the headers and footnotes. An incomplete list of these:
· What are the quantities in the last three lines of table I?
· In table 2, for columns XPS, LOBA, EOS, OSLO a multicolumn stating that these are oxidation state descriptions should be given; for the first two columns some hint is necessary that these are from the ALMO-EDA analysis; Delta CT should be defined here not just in the text.
· In table 2, definition for color coding should be given; reference to the text is not appropriate, since explicit definition is not given there, either (green: OK, yellow: “boarderline”???, orange: “final group”)
· In table 3, the quantity described with “rho at…” should be defined in the header since its relation to the QTAIM is not obvious thus it is hard to find the data when reading the text. The same is true for the following line.
In addition, the interpretation of the data in the tables are made hard by splitting the descriptions in the first rows/columns. Just to give an example: in Table one “M:Sum of angles” is split into two, suggesting the that the first and second row describes different quantities.
Table 1 includes too many data, it is almost unreadable. Since most of the data are not discussed in the text, a large part could be transferred to Supporting materials.
In this paper a large number of methods/tools are used to interpret structure and bonding properties. A section shortly summarizing, or at least defining these could help the reader. This way one can avoid such problems, like the late definition of the acronym QTAIM, etc. Also, it would make a chance to discuss the difference between ALMO-EDA and QTAIM charge transfer values which differ by one order of magnitude thus their combined use does not make a strong case.
It is also hard to distinguish new results and previously obtained one. At some places there are too many emphases on the latter: for example, the first three paragraphs of section 3.6 could be considerably shortened. Even more problematic is Section 5 where I was unable to identify what is new. As the closing sentence of the paper indicates (“We are currently in the final stages of our investigation on the hydrogenation…”) that at most preliminary results are presented here. (This is why I suggest skipping this part.)
In my opinion, the structure and content of section 4 is very strange. The title promises a discussion of the M-X-R rotational barrier. Then (starting with number 2!!) other processes are discussed. In my opinion, this section needs renaming and restructuring.
Some less relevant remarks:
· lines 282-283: occupied-virtual
· lines 297-298: should be made clear which value belongs to what quantity.
· me- is not defined
· line 318: what does “balanced charged transfer” mean? I think it is simply “no charge transfer”.
· line 480: what do you mean on “comparing the Rh, Ir-O and Ir-S bond lengths”?
· line 1033: “reducing destabilizing”
Conclusion is very short and does not touch all the topics the paper discusses. This remark is relevant only, if the authors (and the editor) decides that the paper should be held in one.
The content of the Supplementary material is not described as required (line 1045).
In summary, the paper includes interesting inorganic chemistry fitting into the special issue dedicated to Liebig’s legacy. However, in the present form the paper is not worthy of honoring of our great forefather.
See above.
Reviewer 2 Report
This paper elucidates the syntheses and non-innocent properties of Ir(PDI) and Rh(PDI) complexes incorporating an O,S donor ligand. The research has been thoroughly investigated through a synergy of spectroscopic experiments and comprehensive theoretical calculations. The conclusions drawn from this study are well-founded and strongly supported by both experimental and computational results. Nonetheless, certain revisions to this manuscript would enhance its quality. Hence, I propose the publication of this manuscript as a chemistry article, subject to the implementation of the following improvements.
1. Introduction
The introduction exhibits a well-crafted presentation. However, augmenting the elucidation with a scheme, exemplifying the authors' prior research endeavors, as illustrated in scheme 1, would impart lucidity to the subject matter of this paper.
2. Scheme 3
In complex 11, the notation of R3 in superscript form should be adopted.
3. 3.3 ALMO-EDA
The examination of the noninnocent behavior of transition-metal complexes is generally undertaken through ultraviolet-visible (UV-vis) spectroscopy. The UV-vis spectra of these complexes constitute a valuable resource in comprehending the noninnocent characteristics of Rh and Ir(PDI) complexes.
4. L321
The subscript form should be employed for CT in the context of deltaCT.
5. 5.2 Section 'characterization of the trihydrido product'
The sole substantiation for the trihydride structure resides in the 1H-NMR of the trihydrido complex. It would be beneficial for the authors to incorporate the NMR spectrum either within the main text or as supplementary information (SI). Furthermore, an inquiry into the feasibility of acquiring the IR spectrum of the trihydrido complex is warranted. Upon treating the IrH3 complex with D2, the formation of IrD3 is anticipated. The IR bands arising from Ir-H vibrations can be confidently assigned by drawing comparisons with the spectra of IrH3 and IrD3 complexes, thereby serving as crucial evidence substantiating the trihydride structure.
Round 2
Reviewer 1 Report
The manuscript has been improved considerably compared to the original version. The authors responded adequately to my criticism/suggestions. However, there are still some minor points where, in my opinion, some changes are necessary.
1) there are missing references in the method section: please add the proper citations for the functionals used, for Grimme's D3, COSMO, LNO-CCSD(T), basis sets used.
2) "balanced charge transfer": I insist that such term does not exist. If I am wrong, please provide the appropriate reference. I believe that what we have here is a Frankel coupling of two local excitations and no CT at all.
3) lines 403-405: correct English sentence reads: Energetically, binding is dominated overall by donation, which can be readily seen from the aggregated charge transfer energies of -365 kJ/mol and -135 kJ/mol for CT(PDIIrSMe) and CT(IrSMePDI), respectively (Table 2).
4) the unit me- does not correspond to IUPAC nomenclature.
5) "comparing the Rh, Ir-O and Ir-S bond lengths": what type of bond length is present in Rh? You mean Rh-O and/or Rh-S.
After this points are addressed by the authors, the paper can be published.
